# The Arp2/3 complex controls the development of homeostatic microglia

Shima Safaiyan [1]✉, Maximilian Frosch [2], Tom Bickel [3], Gianni Monaco[2], Roie Dvir[1], Christian Madry [3], Lance Fredrick Pahutan Bosch [4], Katrin Kierdorf[4,5], Metello Innocenti [6], Josef Priller[1,7,8], Marco Prinz [2,5] & Tim Lämmermann [9,10]✉

## Abstract

**Microglial dynamics and homeostasis are crucial for maintaining central nervous system (CNS) function. To fulfill their homeostatic functions, microglia develop into ramified cells with highly dynamic cell protrusions. However, the detailed mechanisms underlying this developmental transition are largely unknown. Here, we investigate the role of the Actin-related protein 2/3 (Arp2/3) complex, a critical actin nucleator that controls the formation of actin branches, for the biology of tissue-resident microglia. By conditionally targeting *Arpc4* in mice, we show that Arp2/3 depletion in tissue-resident microglia causes phenotypes beyond previously reported functions in other immune cell types. Our results identify an important role of Arp2/3 for controlling the developmental transition of microglia into cells with ramified morphology, homeostatic gene profile, and surveillance function in the CNS. Together, our results link actin remodeling to microglial maturation and activation, highlighting the Arp2/3 complex as a critical factor for maintaining the plasticity and preventing pathological activation of endogenous microglia.**

**Keywords** Microglia; Arp2/3 Complex; Actin; Myelin Degeneration; TGFβ Signaling
**Subject Categories** Cell Adhesion, Polarity & Cytoskeleton; Immunology; Neuroscience

## Introduction

Microglia are a subset of immune cells residing in the central nervous system (CNS) parenchyma throughout life. Microglial cells are endowed with morphological plasticity, allowing adaptation to pathophysiological changes in their microenvironment. Hence, it is not surprising that the homeostatic and immune functions of microglia depend on their dynamic behavior (Prinz et al, 2019). In the healthy brain, microglial ramified morphology and the continuous movement of their protrusions are essential for baseline surveillance of the microenvironment (Nimmerjahn et al, 2005). Upon tissue injuries, microglial morphological remodeling from a ramified to an ameboid shape facilitates migration and phagocytosis (Vidal-Itriago et al, 2022). Microglia play a critical role in maintaining CNS homeostasis throughout development and aging (Butovsky and Weiner, 2018; Gomez-Nicola and Perry, 2015; Kettenmann et al, 2011; Safaiyan et al, 2016; Frosch et al, 2025). Dysfunctional microglia are characterized by proinflammatory responses and disrupted functions, such as impaired motility, migration, and phagocytic activity, and they are often associated with brain disorders, including neurodevelopmental and age-related neurodegenerative diseases (Safaiyan et al, 2021; Safaiyan et al, 2016; Wendimu and Hooks, 2022). Yet, the mechanisms controlling the dynamic behavior and functions of microglial cells in the brain are largely unknown. In this regard, actin dynamics drive morphological remodeling and motility in immune cells, including microglia (Franco-Bocanegra et al, 2019). The organization and dynamics of the actin cytoskeleton are controlled by a large array of actin-binding proteins (ABPs), which are involved in the regulation of actin filament assembly and disassembly, as well as actin filament arrangement (Lappalainen, 2016; Pollard, 2016). While many ABPs are commonly expressed in eukaryotic cells, it is also known that distinct ABP isoforms or expression levels fine-tune actin filament dynamics and organize actin networks in a cell-type-specific manner (Lappalainen, 2016). Transcriptomic analyses of mice, monkeys, and humans have identified 45 genes associated with the "actin-binding" pathway that are enriched in microglia (Drew et al, 2020). Many of these genes likely play important roles in maintaining microglial integrity and function in both healthy and diseased states (Galatro et al, 2017; Hammond et al, 2019; Keren-Shaul et al, 2017; Masuda et al, 2019; Van Hove et al, 2019; Ximerakis et al, 2019). However, the functional significance of only a few of these genes, e.g., *Cfl1* (encoding Cofilin-1) and *Cyfip1*

[1]Technical University of Munich, School of Medicine and Health, Department of Psychiatry and Psychotherapy and German Center for Mental Health (DZPG), Munich, Germany. [2]Institute of Neuropathology, Medical Faculty, University of Freiburg, Freiburg, Germany. [3]Institute of Neurophysiology, Charité – Universitätsmedizin Berlin, corporate member of Freie Universität Berlin and Humboldt-Universität zu Berlin, Berlin, Germany. [4]Institute for Infection Prevention and Control, Medical Faculty, University of Freiburg, Freiburg, Germany. [5]Centre for Integrative Biological Signalling Studies (CIBSS), University of Freiburg, Freiburg, Germany. [6]Department of Biotechnology and Biosciences, University of Milano-Bicocca, Piazza della Scienza 2, 20126 Milan, Italy. [7]University of Edinburgh and UK DRI, Edinburgh, UK. [8]Neuropsychiatry and Laboratory of Molecular Psychiatry, Charité Universitätsmedizin Berlin and DZNE, Berlin, Germany. [9]Max Planck Institute of Immunobiology and Epigenetics, Freiburg, Germany. [10]Institute of Medical Biochemistry, Center for Molecular Biology of Inflammation (ZMBE), University of Münster, 48149 Münster, Germany. ✉E-mail: shima.safaiyan@tum.de; laemmermann@uni-muenster.de

(encoding Cytoplasmic FMR1 interacting protein 1 (CYFIP1), has been studied in detail (Crux et al, 2024; Drew et al, 2020). Of note, CYFIP1 is part of a macromolecular complex regulating the formation of actin-based protrusions through the activation of the Actin-related protein 2/3 (Arp2/3) complex (Polesskaya et al, 2022), a key regulator of actin filament branching (Campellone and Welch, 2010; Blanchoin et al, 2014; Vinzenz et al, 2012). The Arp2/3 complex comprises seven subunits (ARP2, ARP3, ARPC1A/B, ARPC2, ARPC3, ARPC4, and ARPC5 (Welch et al, 1997)), of which ARPC2 and ARPC4 ensure the complex's stability and function (Gournier et al, 2001; Rouiller et al, 2008). Interestingly, comparisons between young and aged human microglia have shown that transcript levels of *Arpc1a* and *Arpc1b*, components of the seven-subunit Arp2/3 complex, are downregulated during aging (Galatro et al, 2017). Given that microglial dynamics are dampened with age (Antignano et al, 2023), the age-related downregulation of Arp2/3 subunits suggests that actin filament branching contributes to maintaining microglial functionality. Previous studies on T cells, neutrophils, dendritic cells, and macrophages linked Arp2/3's role in immune cell types primarily to the regulation of cell migration and phagocytosis (Georgantzoglou et al, 2022; Graziano et al, 2019; Kempers et al, 2021; Leithner et al, 2016; Moreau et al, 2015; Obeidy et al, 2020; Rotty et al, 2017; Ruef et al, 2023; Tsopoulidis et al, 2019). However, the role of the Arp2/3 complex for tissue-resident microglia and their function in vivo have never been directly addressed. Here, we reveal an unprecedented role of the Arp2/3 complex in the developmental transition of endogenous microglia into a resting ramified state, morphological remodeling for basal motility and chemotaxis, showing that the Arp2/3 complex contributes to the development of microglia homeostasis in the CNS. A companion study by Paulson and co-workers shows that disrupting Arp2/3 function in mature microglia using chemical perturbation leads to alterations in their characteristic branched morphology and compromises their ability to respond to iC3b-mediated haptotactic cues in their native environment (Paulson et al, 2026). Together, these two studies underscore the Arp2/3 complex as a pivotal regulator of microglial development and function, emphasizing its crucial role in both the formation and maintenance of microglial activities in the adult CNS.

## Results

### Arp2/3 complex regulates microglial cell ramification and baseline surveillance

Since we are missing in vitro methods that recapitulate all hallmarks of adult homeostatic microglia development (Timmerman et al, 2018), we targeted the Arp2/3 complex directly in microglia in vivo. To deplete the Arp2/3 complex in microglia developing and residing in the CNS tissue, we targeted the Arp2/3 subunit, ARPC4, which is a critical component of the Arp2/3 complex. Hence, we generated a conditional knockout mouse model, *Tg(Cx3cr1-Cre) Arpc4^fl/fl* (*Arpc4^ΔCx3cr1*), in which the *Arpc4* gene is deleted in brain myeloid cell populations, including microglia, from their onset of development (Mroue-Ruiz et al, 2024; Zhao et al, 2019). Flow cytometric analyses showed efficient depletion of not only ARPC4 but also ARPC2 in microglia isolated

from *Arpc4^ΔCx3cr1* mice in comparison to microglia derived from Cre-negative *Arpc4^fl/fl* littermate controls, indicating that the integrity of the Arp2/3 complex is compromised (Fig. EV1A), as previously observed in other cell types (Kaltenbach et al, 2024; van der Kammen et al, 2017; Zhao et al, 2021). Furthermore, *Arpc4*-deficient microglia showed reduced global F-actin content compared to control microglia, as measured by flow cytometric detection of fluorescent phalloidin (Fig. EV1B). Next, we assessed microglia homeostasis in the CNS by performing immunohistochemistry (IHC) on brain sections. For the identification of microglia in the parenchyma of homeostatic CNS, we used the common marker IBA1, which can also detect other myeloid cells outside the parenchyma. To rule out off-target effects of Cre recombinase expression on microglia phenotypes in the used mouse strain, we initially compared adult mice with and without *Tg(Cx3cr1-Cre)*. Comparisons of age-matched *Tg(Cx3cr1-Cre) Arpc4^+/+* and Cre-negative *Arpc4^fl/fl* mice showed that microglia numbers, ramification, and surveillance index were comparable between both genotypes (Fig. EV1C–E). Additionally, we did not detect any differences in the expression of microglia markers IBA1 and P2RY12 (Fig. EV1F). We also showed that *Tg(Cx3cr1-Cre)* expression does not induce microglia activation (Fig. EV2A–C). Based on these results, we decided to use Cre-negative *Arpc4^fl/fl* mice as control animals for comparisons with age-matched littermate *Arpc4^ΔCx3cr1* mice, allowing us to reduce surplus mouse generation in animal breeding.

To investigate the role of the Arp2/3 complex for microglia in the CNS, we analyzed brain sections from developing and adult *Arpc4^ΔCx3cr1* and aged-matched control mice. Contrasting the highly branched structure of fine cellular processes in control microglia (Figs. 1A and EV1C), *Arpc4*-deficient microglia exhibited altered morphology with retracted and thick cellular branches (Fig. 1A). Morphological analysis using 3DMorph showed a significant decrease in microglial ramification, as well as reduced occupation of cell territories in the cortex and corpus callosum in adult *Arpc4^ΔCx3cr1* mice (Fig. 1A,B). This alteration was detected across different brain regions, including the corpus callosum, cortical layers, striatum, and hippocampus, in 3- and 12-week-old *Arpc4^ΔCx3cr1* mice (Figs. 1C and EV3A), and observed in both male and female animals (Fig. EV3B). In contrast to reports on tissue-resident mast cells and Langerhans cells (Delgado et al, 2024; Kaltenbach et al, 2024), there was no decrease, but rather a sharp increase in microglial cell number in *Arpc4^ΔCx3cr1* mouse brains from early postnatal stages (3 weeks old) into adulthood (12 weeks old) (Fig. 1D).

Next, we also performed two-photon laser scanning microscopy-based live imaging of GFP-expressing microglia in acute brain slices dissected from adult *Arpc4^ΔCx3cr1 Cx3cr1^GFP/+* (*Tg(Cx3cr1-Cre) Arpc4^fl/fl Cx3cr1^GFP/+*) and *Cx3cr1^GFP/+* control (*Arpc4^fl/fl Cx3cr1^GFP/+*) mice (Fig. 2A,B). Visualization of live dynamics confirmed that ramification and surveillance indices of *Arpc4*-deficient microglia were significantly reduced (Fig. 2C,D; Movie EV1). This phenotype in brain tissue was retained when sorting small numbers of microglia from postnatal *Arpc4^ΔCx3cr1* or control mouse brains and culturing them in 3D Matrigel. In contrast to 2D culture conditions, wild-type microglia preserve their ramified morphology and motility pattern in the 3D system, comparable to microglia dynamics in vivo (Fig. EV3C). However,

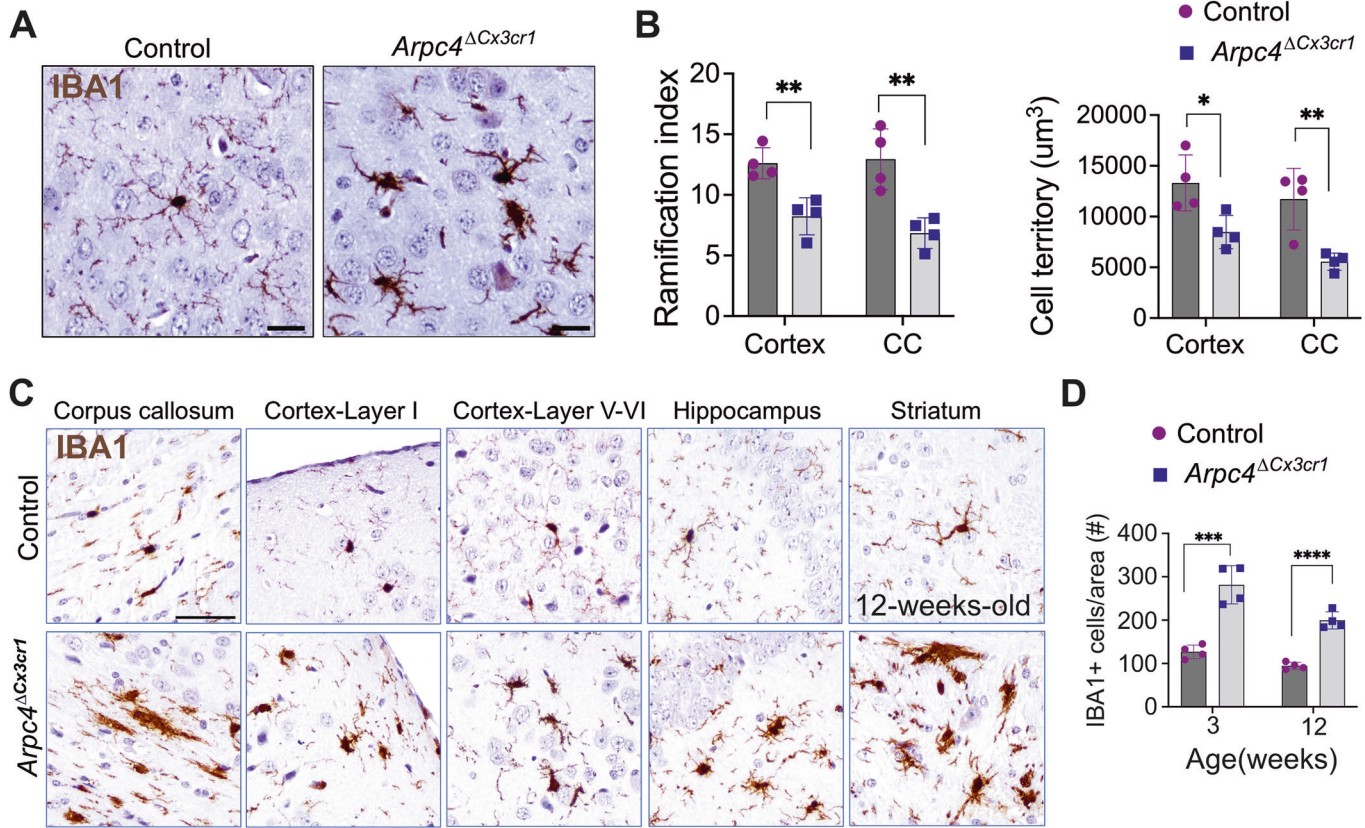

**Figure 1. Microglial morphology is altered in *Arpc4^ΔCx3cr1^* mice.**

(A) Representative DAB-stained images of microglial morphology in 12-week-old *Tg(Cx3cr1-Cre) Arpc4^fl/fl^* (*Arpc4^ΔCx3cr1^*) and *Arpc4^fl/fl^* (control) mice. Microglia were labeled with IBA1 and visualized using DAB (brown). Scale bar: 50 µm. (B) Quantification of microglia ramification and cell territory in 12-week-old *Arpc4^ΔCx3cr1^* and control mice (*n* = 3–4 mice per group, data are presented as mean ± s.d., two-tailed Student's *t*-test, ns not significant; Ramification in cortex: **P = 0.0046; in CC: **P = 0.0049, cell territory in cortex: *P = 0.0237; in CC: **P = 0.0078). (C) Representative DAB-stained images of microglial morphology in different brain regions in 12-week-old *Arpc4^ΔCx3cr1^* and control mice. Microglia were labeled with IBA1 and visualized using DAB (brown). Scale bar: 100 µm. (D) Quantification of microglial cell number in the cortex of 3- and 12-week-old *Arpc4^ΔCx3cr1^* and control mice (*n* = 4 mice per group, data are presented as mean ± s.d., two-tailed Student's *t*-test, in 3-week-old: ***P = 0.0006; in 12-week-old: ****P = 0.00001). Source data are available online for this figure.

3D-cultured *Arpc4*-deficient microglia did not display the typical ramified morphology of control microglia. Instead, these *Arpc4*-deficient cells formed long, thin protrusions that appeared to retract faster compared to the protrusions of the control microglia. In contrast to control cells, *Arpc4*-deficient microglia lost their branches after ten hours of culture and stopped their motility (Fig. EV3D; Movie EV2). Together, these data indicate that the Arp2/3 complex is required for the formation of dynamic cell protrusions associated with the motility of microglial cells and baseline surveillance in vivo.

## Loss of the Arp2/3 complex in microglia results in downregulation of microglia homeostatic gene signatures

To understand whether the observed morphological and migratory phenotypes could be entirely explained by Arp2/3-mediated effects on the cytoskeleton, we also investigated whether *Arpc4*-deficient microglia underwent normal transition from a developing into homeostatic microglia on the transcriptional level. Hence, we

performed bulk RNA sequencing on FACS-based isolated microglia from 12-week-old control and *Arpc4^ΔCx3cr1^* mice. Flow cytometry analysis already revealed a subpopulation of *Arpc4*-deficient microglia exhibiting significantly low levels of CD11b cell surface expression. This subpopulation was absent in control microglia (Fig. 3A). IHC analysis revealed that less than 20% of total IBA1⁺ microglia co-expressed CD11b in *Arpc4^ΔCx3cr1^* mice compared to control mice (Fig. 3B). Thus, we included CD11b^high^ CD45^low^ and CD11b^low^ CD45^low^ cells in the transcriptomics analysis of knockout mice. Pathway analysis using Ingenuity pathway analysis software based on differentially expressed genes (DEGs) showed that actin cytoskeleton signaling, senescence pathway, P2Y purinergic signaling, neuroinflammation signaling, chemokine pathway, and mTOR signaling pathways were differentially regulated in the *Arpc4^ΔCx3cr1^* microglia compared to control cells (Fig. 3C). Further analysis of the DEGs showed that genes involved in the regulation of actin cytoskeleton arrangement were mostly downregulated (*Was, Ssh2, Wasf2, Cdc42, Rac1, Eps8, Apbb1ip, Dnm2, Cyfip1, Marcks, Pdlim4, Fscn1*) (Fig. 3D). *Arpc4*-deficient cells showed upregulation of microglia activation markers, whose expression has already been

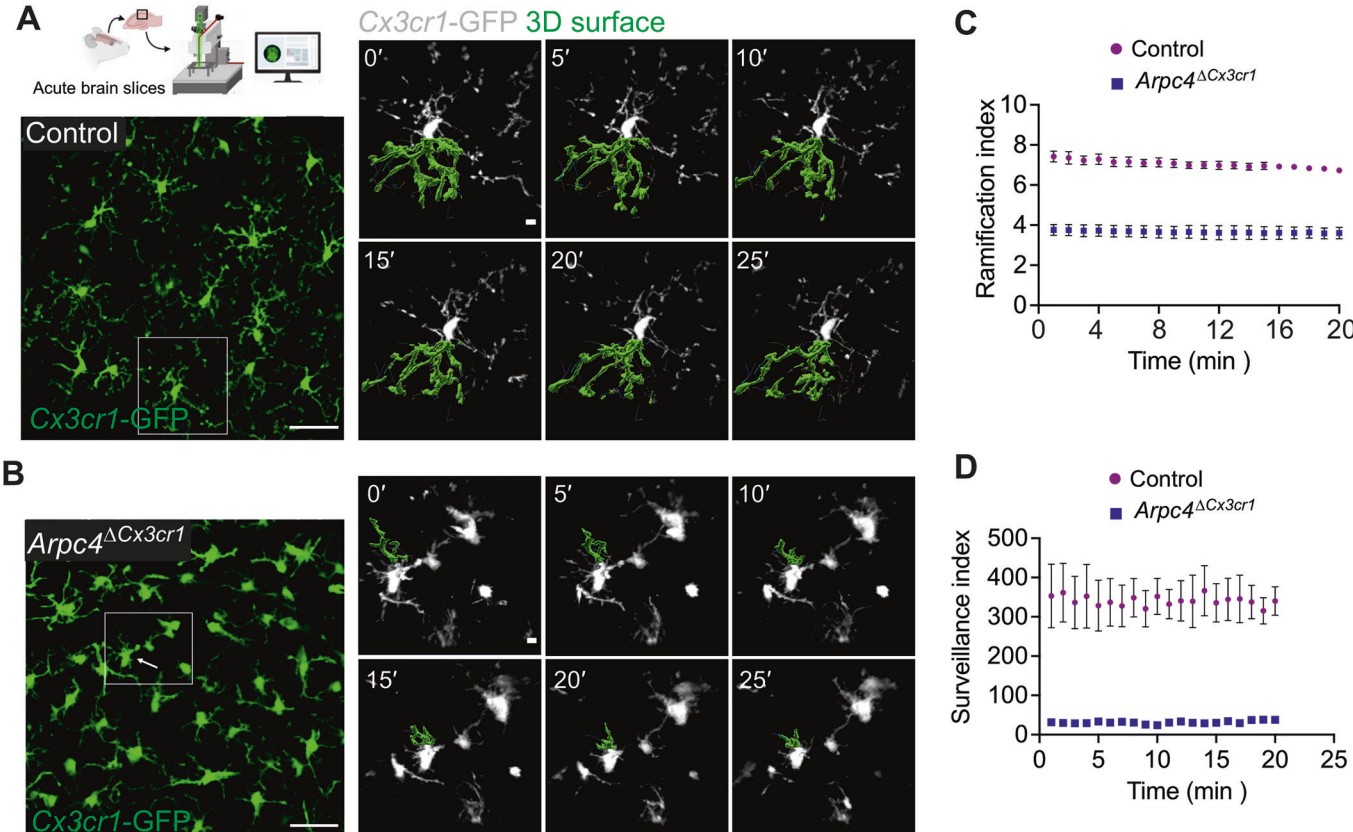

**Figure 2. Microglial motility and baseline surveillance are disturbed in *Arpc4*^ΔCx3cr1^ mice.**

(A, B) Two-photon laser scanning microscopy-based live imaging of microglia in acute brain slices prepared from 12-week-old *Tg(Cx3cr1-Cre) Arpc4*^fl/fl^ (*Arpc4*^ΔCx3cr1^) *Cx3cr1*^GFP/+^ and *Arpc4*^fl/fl^ (control) *Cx3cr1*^GFP/+^ mice. Representative images of time sequences show *Cx3cr1*-GFP signal (white) and one protrusion of a microglia surface-rendered and 3D reconstructed (green) to display microglia protrusion dynamics in both genotypes. Scale bars: 40 µm (overview), 5 µm (zoom-in). (C) Quantification of microglial cell ramification over time in acute brain slices prepared from 12-week-old *Arpc4*^ΔCx3cr1^ *Cx3cr1*^GFP/+^ and control *Cx3cr1*^GFP/+^ mice (*n* = 3 mice per group; data represent mean ± SEM of mouse averages, with 43 cells (Control) and 51 cells (*Arpc4*^ΔCx3cr1^) analyzed in total, Student's *t*-test, *P* = 9.69 × 10^−4^). (D) Quantification of brain area surveyed by a microglial cell per minute (surveillance index) over time in acute brain slices prepared from 12-weeks-old *Arpc4*^ΔCx3cr1^ *Cx3cr1*^GFP/+^ and control *Cx3cr1*^GFP/+^ mice (*n* = 3 mice per group, data were presented as mean ± SEM of three mice; the surveillance indices of 43 cells (Control) and 51 cells (*Arpc4*^ΔCx3cr1^) analyzed in total were analyzed, Welch's *t*-test, *P* = 0.01891). Source data are available online for this figure.

associated with neurodegenerative diseases and brain inflammation (*Axl, Clec7a, Lgals3, Spp1, Lpl, Cst7, Tyrobp*) (Keren-Shaul et al, 2017), and downregulation of homeostatic genes, including purinergic receptors and checkpoint genes (*Cx3cr1, Hexb, Tmem119, P2ry12, P2ry13, Csf1r*) (Fig. 3E). This transcriptomic profiling further revealed that genes related to the TGFβ signaling pathway, including *Tgfb1, Tgfb2, Smad1, Smad2,* and *Smad3*, were differentially expressed (Fig. 3F).

Based on this transcriptomic comparison, we further validated selected DEGs in microglia of control and *Arpc4*^ΔCx3cr1^ mice. IHC analyses confirmed that *Arpc4*-deficient microglia upregulated activation markers such as Galectin-3 and AXL in various brain regions, including the cortex, corpus callosum (CC), and the striatum, during development (3-week-old) and adulthood (12-week-old) (Fig. 3G,H). Additionally, lysosomal markers, such as CD68, which potentially indicate lysosomal activity, were upregulated on the *Arpc4*-deficient microglia (Fig. EV3E). The downregulation of other microglial homeostatic markers, such as Transmembrane Protein 119 (TMEM119), was confirmed using

flow cytometric analysis (Fig. 3I). Thus, our results clearly show that *Arpc4*-deficiency in tissue-resident microglia not only results in morphological changes but also substantially impacts their transcriptional and protein expression profile.

To confirm one of the most striking findings from the bulk RNAseq analysis, we performed IHC analysis of P2RY12, a purinergic chemotactic receptor that is critical for tissue surveillance and the response to injury and tissue surveillance (Davalos et al, 2005). Immunofluorescence in adult brain tissues revealed an almost complete absence of P2RY12 expression in microglia of adult *Arpc4*^ΔCx3cr1^ mice compared to control mice (Figs. 4A and EV1F). To test the functional consequence of this cellular phenotype, we visualized microglia with two-photon laser microscopy in brain slices of adult *Arpc4*^ΔCx3cr1^ *Cx3cr1*^GFP/+^ or control *Cx3cr1*^GFP/+^ mice and followed their dynamic response to a laser-induced tissue damage. While control microglia formed pronounced protrusions, which extended chemotactically toward the injury site, *Arpc4*-deficient microglia were almost completely blunted in their chemotactic response (Fig. 4B,C; Movie EV3). Since a previous report by Drew

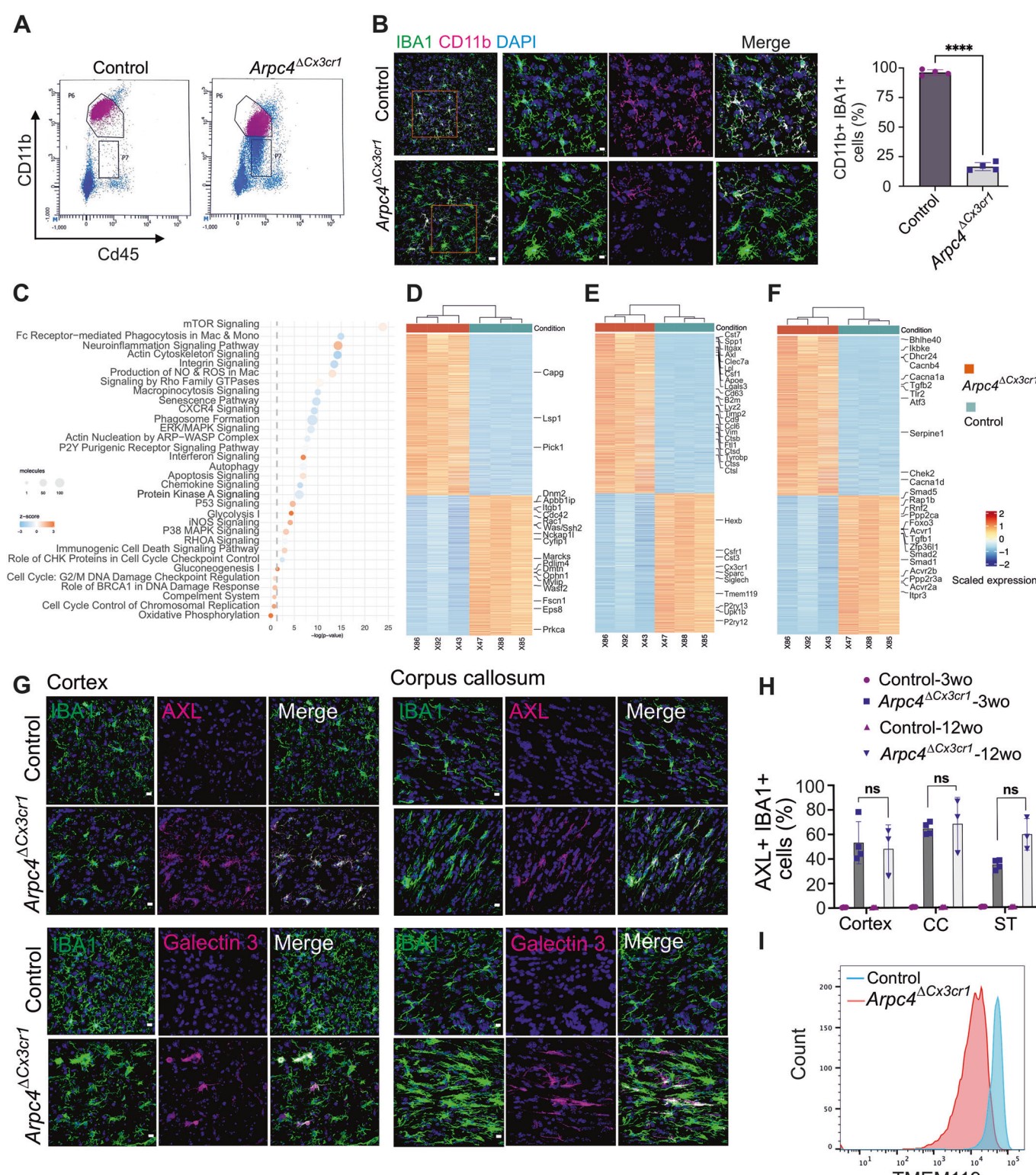

et al (2020) showed that acute Arp2/3 inhibition by CK666 treatment hardly impaired the chemotactic response of microglia in this experimental setup, we conclude that the drastic downregulation of purinergic receptors underlies the impaired response of Arp2/3-depleted microglia. Together, our data show that the Arp2/3 complex induces transcriptional and cellular changes beyond direct cytoskeletal control, which influence the functional responses of microglia in vivo. Thus, the Arp2/3 complex is a cell-intrinsic contributor to homeostatic gene expression and tissue surveillance in CNS microglia.

◄ **Figure 3.   Microglia gene signature is downregulated in *Arpc4^ΔCx3cr1* mice.**

(A) Flow cytometric analysis and dot plots showing the microglial subpopulations (CD45^low CD11b^high and CD45^low CD11b^low) in 12-week-old *Tg(Cx3cr1-Cre) Arpc4^fl/fl* (*Arpc4^ΔCx3cr1*) and *Arpc4^fl/fl* (control) mice (Biological replicates: $n = 6$ mice per group, Technical replicates: $n = 4$ experiments). (B) Representative images and quantification of CD11b^+ microglia in 12-week-old *Arpc4^ΔCx3cr1* and control mice, scale bar: 20 μm ($n = 4$ mice per group, data are presented as mean ± s.d., two-tailed Student's *t*-test; ****$P = 0.00001$). (C) Pathway analysis using Ingenuity pathway analysis software (IPA, QIAGEN) based on the differentially expressed genes (DEGs) (adj. *P* value <0.05, fold change >2). (D–F) Heatmaps of selected pathways show the average expression of DEGs related to actin cytoskeleton (D), cell activation and homeostasis (E), and senescence pathway (F). (G) Representative images of AXL and Galectin-3 expression in IBA1^+ microglia in the cortex, corpus callosum in 12-week-old *Arpc4^ΔCx3cr1* and control mice. Scale bar: 10 μm. (H) Quantification of AXL expression in IBA1^+ microglia in the cortex, corpus callosum (CC), and striatum (ST) in 3- and 12-weeks-old *Arpc4^ΔCx3cr1* and control mice ($n = 3$–4 mice per group, data were presented as mean ± s.d, two-way ANOVA followed by Bonferroni's post hoc test, AXL^+ microglia in cortex, CC, and ST in 3-weeks-old compared to 12-weeks-old: ns: $P > 0.9999$). (I) Flow cytometric analysis of TMEM119 expression in 12-week-old *Arpc4^ΔCx3cr1* and control mice (Biological replicates: $n = 4$ mice per group, Technical replicates: $n = 2$ experiments). Source data are available online for this figure.

## Loss of the Arp2/3 complex induces a reactive phenotype in microglia, connected with enhanced phagocytosis of myelin

Since bulk RNA sequencing lacks the fine-grained analysis to characterize a potentially heterogeneous cell population, we also performed single-cell RNA sequencing. In particular, we were interested in whether the observed phenotypic changes in *Arpc4^ΔCx3cr1* mice were connected to a specific microglial state or subtype. After quality control, a total of 14,752 nuclei were analyzed from FACS-based isolated CD45^+ cells of 12-week-old control and *Arpc4^ΔCx3cr1* mice. Cell type assignment was conducted using published gene sets (Jordão et al, 2019), identifying different types of immune cells, including microglia, CNS-associated macrophages (CAMs), monocytes, T cells, and others (Figs. 5A,B and EV4A). We then focused on microglia in our dataset, in which six distinct microglia clusters (C1-6) could be identified (Fig. 5C). Control and *Arpc4*-deficient microglia clustered separately based on their transcriptomic profile. While C1, C2, and C3 comprised mostly microglia from control mice, C0, C4, and C5 were clearly expanded in *Arpc4^ΔCx3cr1* mice (Fig. 5C,D). Differentially expressed gene (DEG) analysis showed that all three microglia clusters from *Arpc4^ΔCx3cr1* mice shared the downregulation of homeostatic signature-associated genes, such as *Siglech, Hexb, P2ry12, Cx3cr1*, and *Tmem119* (Fig. EV4B,C). Cluster-specific gene analysis highlighted Cluster 4 (C4), which was characterized by elevated gene expression of *Apoe* and *Ms4a7* (Figs. 5E and EV4C). Of note, C4 also exhibited a lower expression level of *Itgam* (encoding CD11b) compared to the other *Arpc4^ΔCx3cr1* microglia clusters (C0 and C5) and homeostatic clusters C1-3 (Fig. EV4D). To determine the spatial localization of C4 microglia, we co-stained IBA1^+ microglia with the RNA in situ probe against *Ms4a7* by combining IHC and RNAscope. This analysis confirmed the colocalization of *Ms4a7* RNA with IBA1-immunoreactive puncta, indicating expression within microglia (Fig. 5F). The *Ms4a7^+* microglia were distributed almost all over the brain (Fig. 5G). These data suggest that, in addition to reduced cell dynamics and impaired homeostasis, the microglial subpopulation C4 displays a reactive phenotype in *Arpc4^ΔCx3cr1* mice. Next, we wondered whether the detected reactive microglia in *Arpc4^ΔCx3cr1* mice show altered functional characteristics during CNS homeostasis. Recent work has shown that homeostatic microglia are important for preventing the degradation of myelin, which is required for the function of neuronal axons, in the adult CNS (McNamara et al, 2023). Hence, we performed IHC to determine the functional interaction of microglia with myelin tracks, which were labeled with the antibody against myelin basic protein (MBP). Our analysis revealed that reactive IBA1^+ microglia upregulated the lysosomal marker CD68 in the vicinity of myelin tracks and engulfed MBP-positive myelin debris in *Arpc4^ΔCx3cr1* mice, which we could hardly observe in control mice (Fig. 5H). Together, our results show that *Arpc4* deficiency causes a loss of core homeostatic gene signatures in adult microglia and the acquisition of a reactive phenotype that interferes with the maintenance of myelin integrity.

## Arp2/3 loss of function correlates with impaired TGFβ signaling in microglia

Lastly, we also followed up on the differential expression pattern of genes related to the Transforming growth factor-β (TGFβ) signaling pathway (Fig. 3F). TGFβ signaling is a master regulator of microglial cell morphology and homeostatic functions (Bureta et al, 2019; Spittau et al, 2020; Zöller et al, 2018). Interestingly, our comparison of transcriptional data revealed that microglia in *Arpc4^ΔCx3cr1* mice share the gene signature of *Tgfbr2*-deficient microglia (Fig. 6A), which have previously been shown to have an active phenotype and express priming markers (Zöller et al, 2018). In particular, the sequencing cluster C4 of *Arpc4*-deficient microglia (Fig. 6B) exhibited a reduced expression profile of TGFβ-dependent homeostatic genes, including *P2ry12, Tmem119, Mef2a, Sall1, Olfml3, and Fcrls* (Butovsky et al, 2014), compared to other microglial clusters in both control and *Arpc4^ΔCx3cr1* mice (Fig. 6B). By performing REACTOME pathway analysis with the 1000 most significantly enriched genes in C4, we found that TGFβ-related pathways appeared among the most significantly altered pathways (Fig. 6C). Given these striking similarities between *Tgfbr2*- and *Arpc4*-deficient adult microglia, we aimed to address whether TGF-β signaling is altered in *Arpc4*-deficient microglia. To do so, we analyzed the phosphorylated versions of SMAD2 and SMAD3 (pSMAD2 and pSMAD3), which are translocated into the cell nucleus in response to TGFβ receptor signaling. When performing IHC for these molecules on brain sections, we detected them in IBA1^+ (microglia) and IBA1^- cells. When focusing on microglia, we observed that nuclear pSMAD signals were detectable throughout microglia in control mice. However, the percentage of microglia that were lacking nuclear pSMAD2 and pSMAD3 immunoreactivity were substantially increased in *Arpc4^ΔCx3cr1* mice (Fig. 6D). Thus, these data correlate the lack of Arp2/3 complex in microglia with impaired TGF-β signaling, an important pathway that maintains microglia development and homeostasis.

Together, we conclude that the Arp2/3 complex is critical for establishing a homeostatic microglia phenotype on the morphological and gene expression level, which is important to preserve a healthy CNS.

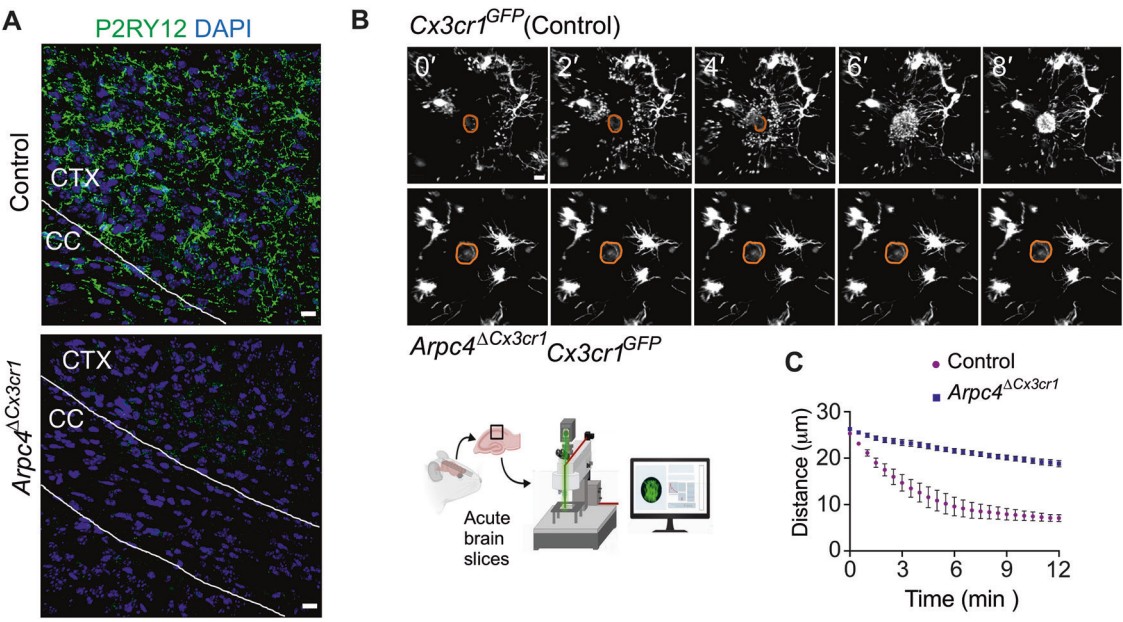

**Figure 4. *Arpc4*-deficiency impairs microglial chemotaxis.**

(A) Representative images from static confocal microscopy displaying P2RY12 expression levels in 12-week-old *Tg(Cx3cr1-Cre) Arpc4^fl/fl^* (*Arpc4^ΔCx3cr1^*) and *Arpc4^fl/fl^* (control) mice. Scale bar: 15 μm. (B) Representative images of two-photon laser scanning microscopy-based live imaging in acute brain slices from 12-week-old *Arpc4^ΔCx3cr1^ Cx3cr1^GFP/+^* and *Cx3cr1^GFP/+^* control mice showing microglial chemotaxis toward the laser-damaged spot (in the orange dashed circle). Scale bar: 10 μm. (C) Time course of directed motility quantified as the distance of microglial processes to the damaged spot (*n* = 3 mice per group; 27 control and 33 *Arpc4^ΔCx3cr1^* chemotactic events analyzed in total, averaged per mouse for statistical comparison; data shown as mean ± SEM of mouse averages, Student's t-test (unpaired), *P* = 0.025). Source data are available online for this figure.

## Discussion

Our study presents a previously unrecognized role of the Arp2/3 complex in the development of microglia homeostasis in the CNS. Genetic targeting of the critical subunit *Arpc4* allowed us to study the functional role of the Arp2/3 complex in microglia, a tissue-resident immune cell type that is difficult to study outside its physiological environments. Our findings contribute to the emerging topic that the depletion of the Arp2/3 complex and interference with Arp2/3 function and actin branching cause distinct cellular phenotypes depending on the studied immune cell type. Initial studies on Arp2/3 complex function in immune cells have largely focused on the analysis of migration and phagocytosis properties due to Arp2/3's key role in actin branching during lamellipodia and phagocytic cup formation. However, studies across different immune cell types now clearly show that the Arp2/3 complex supports diverse immune functions. These comprise phagocytosis and integrin signaling in macrophages (May et al, 2000; Rotty et al, 2017b), polarity, motility, transendothelial migration, and environmental exploration in leukocytes (Graziano et al, 2019; Leithner et al, 2016a, Peterson et al, 2024; Kempers et al, 2021; Vasconcellos et al, 2025), dendritic cell trafficking (Vargas et al, 2016), T cell homeostasis (Obeidy et al, 2020; Zhang et al, 2017) granule exocytosis of NK cells (Carisey et al, 2018). Recent studies have revealed isoform diversity among Arp subunits (Abella et al, 2016; Sadhu et al, 2023) and cell-type-specific regulatory mechanisms (Franco-Bocanegra et al, 2019),

which likely contribute to the various functional roles of the Arp2/3 complex in distinct immune cell types.

The genetic defects in the ARPC1B and ARPC5 isoforms in humans are associated with immune system disorders, highlighting the clinical relevance of Arp2/3 complex dysfunction (Gamze Sonmez et al, 2025; Kuijpers et al, 2017; Nunes-Santos et al, 2023; Sindram et al, 2023). In mice, only a few studies have directly targeted Arp subunits in distinct endogenous immune cell types to understand cell-type-specific Arp2/3 functions in the physiological tissue setting. However, recent studies in mice have targeted the *Arpc4* subunit in endogenous mast cells or Langerhans cells and highlighted a critical role of the Arp2/3 complex for the survival of tissue-resident immune cells (Delgado et al, 2024; Kaltenbach et al, 2024). Additionally, loss of *Arpc5* in the murine hematopoietic system alters the functions of mononuclear phagocytes and causes early-onset intestinal inflammation (Vasconcellos et al, 2025).

We here extend the knowledge on the physiological functions of the Arp2/3 complex to the CNS and another tissue-resident immune cell type. By targeting *Arpc4* in tissue-resident microglia in mice, we identify a critical role for Arp2/3 function in microglial development and homeostasis in the CNS. In the normally developing brain, up to the second postnatal week, immature microglia appear with an ameboid shape and enhanced lysosomal activity, expressing genes related to an active state. As microglia mature, they shift from a rounded morphology to a more ramified form characterized by increasing protrusion formation. Our findings demonstrate that *Arpc4*-deficient microglia, which lack

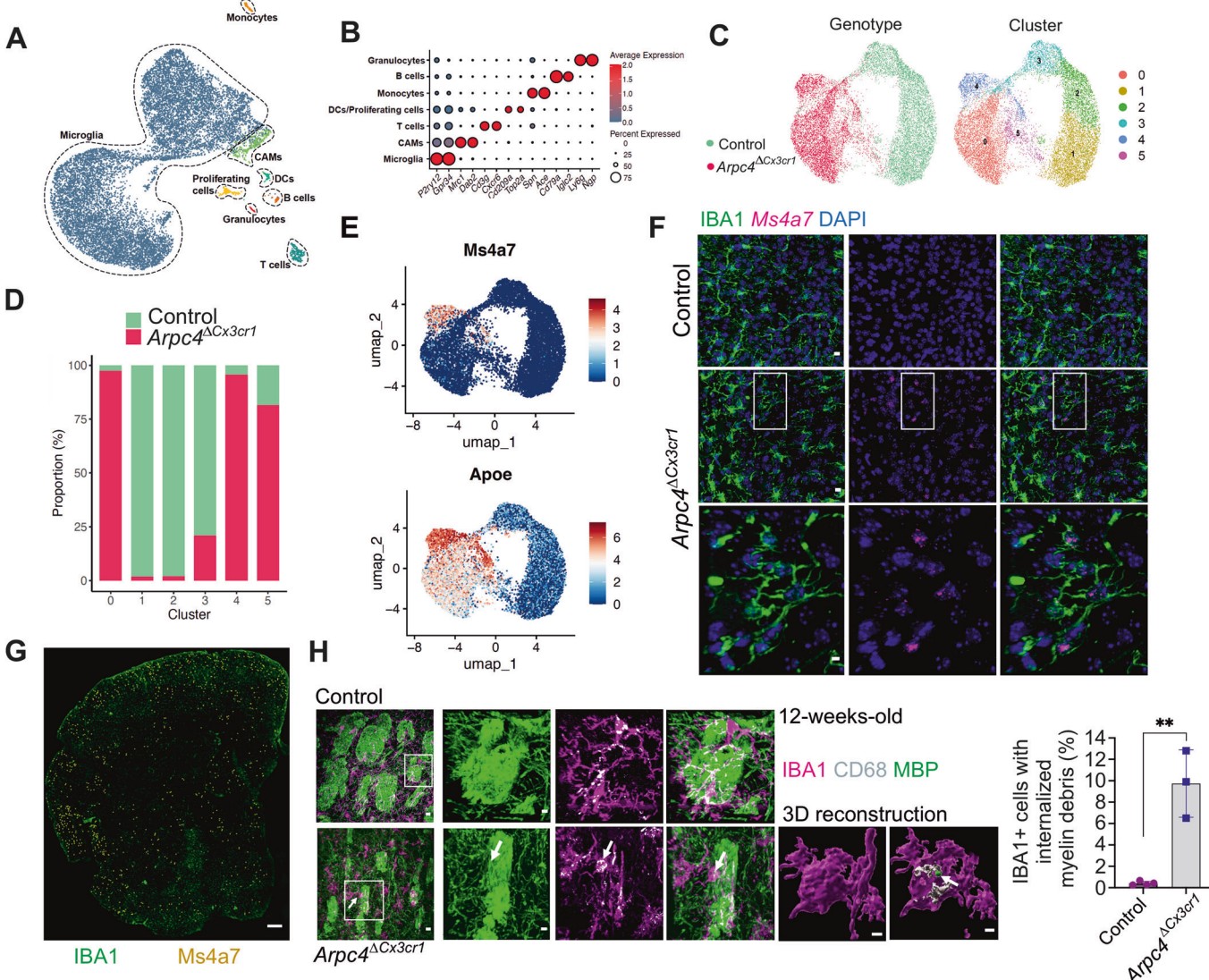

**Figure 5. *Arpc4*-deficient microglia lose their homeostatic core gene signature and acquire a reactive phenotype.**

(A) Uniform manifold approximation and projection (UMAP) visualization of 14,752 individual cells captured by scRNAseq from 12-week-old *Tg(Cx3cr1-Cre) Arpc4fl/fl* (*Arpc4ΔCx3cr1*) and *Arpc4fl/fl* (control) mice. (B) Selected marker genes associated with each cell type are highlighted in (A). (C) UMAP visualization of microglial cells, highlighting clusters and genotypes. (D) Marimekko chart of microglia depicting the proportions of *Arpc4ΔCx3cr1* and control microglia for each cluster. (E) Expression levels of *Apoe* and *Ms4a7*. (F) Representative images showing the colocalization of *Ms4a7* RNA with IBA1-immunoreactive puncta in 12-week-old *Arpc4ΔCx3cr1* and control mice. Scale bars: 10 μm (overview), 5 μm (zoom-in). (G) An overview confocal image showing the colocalization of *Ms4a7* RNA with IBA1+ cells all over the brain in 12-week-old *Arpc4ΔCx3cr1* mice. Scale bar: 100 μm. (H) Representative images and quantification of IBA1+ cells with internalized myelin debris (MBP+) in 12-week-old *Arpc4ΔCx3cr1* and control mice. The 3D reconstruction shows internalized myelin fragments, associated with the lysosome (CD68+) within microglia. Scale bars: 10 μm (overview), 4 μm (zoom-in), 2 μm (3D) (n = 3–4 mice per group, data are presented as mean ± s.d, two-tailed Student's *t*-test, **P = 0.0017. Source data are available online for this figure.

Arp2/3-mediated actin branching, are unable to adopt a ramified morphology within CNS tissue and remain in their immature, ameboid state. This lack of morphological transition in *Arpc4*-deficient microglia coincides with the downregulation of resting and homeostatic genes, which are normally expressed when microglia transform toward the ramified state (Dermitzakis et al, 2023; Lull and Block, 2010; Prinz et al, 2019; Wurm et al, 2021). Hence, the Arp2/3 complex is critical for controlling the establishment of a homeostatic microglia state on the morphological and transcriptional level. The study by Paulson et al underscores the crucial importance of the Arp2/3 complex in preserving the

intricate, branched morphology of mature microglia, as demonstrated by experiments using CK666-mediated inhibition of Arp2/3 function in hippocampal slice cultures (Paulson et al, 2026).

Our study highlights that targeted depletion of the Arp2/3 complex in endogenous immune cells can result in transcriptional and expression changes of several cellular pathways, which influence the cellular phenotype in the tissue, but are not directly connected to Arp2/3's most known role as actin nucleator. Such functions beyond actin cytoskeleton control have largely been neglected in previous studies of the Arp2/3 complex in other immune cell types. Strikingly, the phenotypic changes of *Arpc4*-deficient microglia also involved a

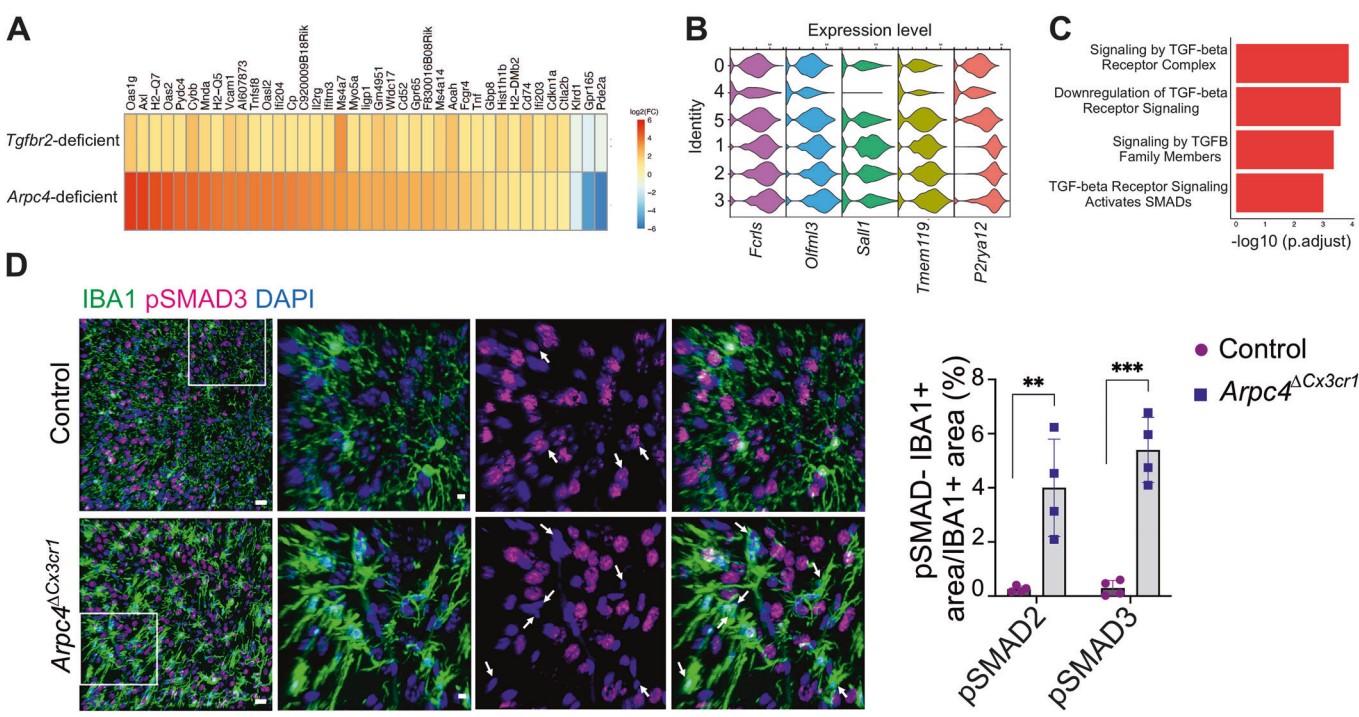

**Figure 6. Arpc4-deficient microglia exhibit impaired TGFβ signaling.**

(A) Comparison of gene profiles in *Tgfbr2-* and *Arpc4*-deficient microglia. (B) Expression levels of common microglial homeostatic genes in microglia clusters. (C) Significantly altered TGF-β signaling-related REACTOME Pathways (Adj. *p* < 0.05). (D) Representative images showing nuclear immunolabeling of pSMAD3 and DAPI in IBA1⁺ microglia, quantification of the pSMAD2⁺ and pSMAD3⁺ signal within IBA1⁺ area in 12-week-old *Arpc4^ΔCx3cr1^* and control mice (*n* = 4 mice per group, data are presented as mean ± s.d., two-tailed Student's *t*-test, pSMAD2: **P = 0.0059; pSMAD3: ***P = 0.0002). Source data are available online for this figure.

transcriptional downregulation of purinergic receptors and a lack of P2RY12 expression, which is critical to sense ATP released from neurons (Lin et al, 2020). In the presence of a tissue injury, microglia switch from random to directed chemotactic motility (Hines et al, 2009; Madry and Attwell, 2015), which is mediated by morphological remodeling and microglial sensome (Das and Chinnathambi, 2020; Hickman and El Khoury, 2019). Our data show that *Arpc4*-deficient microglia exhibit defective migration toward a site of CNS injury, which is likely due to impaired purinergic receptor signaling. Our conclusion is supported by studies from Paulson and co-authors (Paulson et al, 2026) and Drew et al, 2020, which demonstrate that pharmacological inhibition of Arp2/3 function in mature microglia, displaying intact purinergic receptor signaling, does not compromise the chemotactic response to ATP released from a micropipette.

When inflammatory stimuli persist, WT microglia are fully activated with enlarged cell bodies and short and thick protrusions and migrate efficiently (Kettenmann et al, 2011), accompanied by changes in gene expression (Lull and Block, 2010). After tissue injury is properly repaired, microglia typically return to their homeostatic state (Miller and Streit, 2007). However, persistent activation can occur, leading to a "primed" state (Candlish and Hefendehl, 2021; Olah et al, 2018; Safaiyan et al, 2021). Primed microglia exhibit an exaggerated and prolonged proinflammatory response that can be detrimental to brain tissue (Niraula et al, 2017). Our findings showed that *Arpc4*-deficient microglia undergo significant morphological and functional changes, which are associated with the loss of their homeostatic gene signature and the upregulation of genes linked to microglial activation.

In this regard, we identified a specific state of reactive microglia characterized by the upregulation of *Apoe* and Membrane-spanning 4-domains, subfamily A (*Ms4a*) family genes. Interestingly, *Ms4a* family members, such as *Ms4a7* and *Ms4a6c*, regulate immune cell function (Eon Kuek et al, 2016) and are associated with Alzheimer's disease (AD) risk (Hollingworth et al, 2011; Ma et al, 2015; Naj et al, 2011), and the APOE pathway is known to drive microglial activation and contribute to the development of a neurodegenerative microglial phenotype (MGnd) (Krasemann et al, 2017; Shi et al, 2019). Most importantly, the reactive *Arpc4*-deficient microglia appear to influence myelin integrity through adulthood, which could be a risk of neurodegeneration (Depp et al, 2023; Nasrabady et al, 2018; Parrilla et al, 2024). Antagonism between cell migration and macro/phagocytosis has previously been described for other immune cell types (Edwards-Jorquera et al, 2020). Upregulated expression of some members of the formin/diaphanous family of proteins in *Arpc4*-deficient microglia (Fig. EV5) may underly the observed increased uptake of MBP extracellular material.

To date, the functional role of the Arp2/3 complex in microglia has not been investigated in the context of brain disease. Our findings align with previous reports showing downregulation of genes related to Arp2/3 subunits in aged microglia in both mice and humans (Galatro et al, 2017; Hammond et al, 2019). Aging microglia are implicated in the progression of neurodegenerative diseases and characterized by impaired chemotaxis, reduced morphological plasticity, and a heightened, primed activation state (Antignano et al, 2023; Norden and Godbout, 2013). These parallels

highlight the Arp2/3 complex as a novel and potentially important factor in CNS disease, encouraging further investigation into its role in microglial dysfunction and neurodegeneration.

Furthermore, our results show a correlation between the Arp2/3 complex and TGFβ signaling, which is essential for microglial homeostasis and maturation and controls their activation (Li et al, 2024; Spittau et al, 2020; Zöller et al, 2018). The fact that nuclear translocation of phosphorylated SMAD2/3 was impaired in *Arpc4*-deficient microglia could provide a mechanistic explanation for many of the observed phenotypes. Of note, SMAD complex activation and nuclear translocation depend on full TGFβ receptor activation, internalization, and trafficking (Chen, 2009; Huang and Chen, 2012), which are regulated by several actin regulatory proteins (Melchionna et al, 2021; Moustakas and Heldin, 2008; Liu et al, 2016). Notably, the Arp2/3 complex has also been implicated in actin-based clathrin-mediated endocytosis and trafficking (Kaksonen et al, 2006; Toshima et al, 2005). Further investigation is required to determine the specific role of the Arp2/3 complex in the TGFβ signaling pathway in microglia.

In summary, our study provides novel insight into the contribution of the Arp2/3 complex to the homeostasis of microglia, the major resident immune cell type in the CNS. We highlight an important role of the Arp2/3 complex for mediating the developmental transition into microglia with ramified morphology, homeostatic gene expression, and tissue surveillance function. Given the multiple roles of the Arp2/3 complex in many cellular processes, including cytosolic and nuclear functions, future studies are needed to determine the detailed molecular mechanism underlying the observed tissue phenotype, which is difficult to dissect in endogenous microglia. However, our findings identify the Arp2/3 complex as a critical determinant for establishing microglia homeostasis, expanding the functional spectrum of this protein complex in the immune system. Together, our study and the accompanying report by Paulson et al, (2026), identify the Arp2/3 complex as a key regulatory component governing microglial development and function within the adult CNS.

# Methods

### Reagents and tools table

| Reagent/resource | Reference or source | Identifier or catalog number |
|---|---|---|
| **Experimental models** | | |
| C57BL/6J wild-type mice | Charles River | |
| Arpc4flox mice | Prof. Metello Innocenti (Università degli Studi di Milano-Bicocca) | van der Kammen et al, 2017 |
| Tg(Cx3cr1-Cre) MW126Gsat/Mmucd mice | MMRRC | Stock #036395-UCD |
| Cx3cr1GFP knock-in mice | Jackson Laboratory | Strain No. 005582 |
| **Antibodies** | | |
| Rabbit anti-IBA1 | Wako | Cat#019-19741 |
| Guinea pig anti-IBA1 | Synaptic Systems | Cat#234 308 |

| Reagent/resource | Reference or source | Identifier or catalog number |
|---|---|---|
| Rat anti-MAC2 (Galectin-3) | BioLegend | Cat#125402 |
| Chicken anti-MBP | Thermo Fisher Scientific | Cat#PA1-10008 |
| Rat anti-CD68 | Bio-Rad | Cat#MCA1957 |
| Goat anti-AXL | R&D Systems | Cat#AF854 |
| Rat anti-P2RY12 | BioLegend | Cat#848002 |
| Rabbit anti-Phospho-SMAD2 (Ser465/467) | Merck | Cat#AB3849-I |
| Rabbit anti-Phospho-SMAD3 (Ser423/425) | Thermo Fisher Scientific | Cat#44-246 G |
| Rat anti-TMEM119 (PE, V3RT1Gosz) | Invitrogen | Cat#12-6119-82 |
| Mouse anti-CD45 (APC, 104) | Thermo Fisher Scientific | Cat#17-0454-82; RRID: AB_469400 |
| Rat anti-CD11b (PE, M1/70) | Thermo Fisher Scientific | Cat#25-0112-82 |
| Rat anti-CD3 (APC/Cy7, 17/A2) | BioLegend | Cat#100222 |
| Rat anti-Ly6G (APC/Cy7, 1A8) | BioLegend | Cat#127624 |
| Rat anti-CD19 (APC/Cy7, 6D5) | BioLegend | Cat#115530 |
| Rat anti-Ly6C (APC/Cy7, HK1.4) | BioLegend | Cat#128025 |
| **Chemicals, enzymes and other reagents** | | |
| Neural Tissue Dissociation Kit (Papain) | Miltenyi Biotec | Cat# 130-092-628 |
| Zombie Violet Fixable Viability Kit | BioLegend | Cat# 423113 |
| Matrigel Basement Membrane Matrix | Corning | Cat# 356234 |
| RNAscope Multiplex Fluorescent Detection Reagents v2 | Advanced Cell Diagnostics (ACD) | Cat# 323100 |
| VECTASTAIN ABC Kit (Standard) | Vector Laboratories | Cat# PK-6100 |
| Tissue-Tek O.C.T. Compound | Sakura Finetek | Cat# 4583 |
| RNeasy Micro Kit | QIAGEN | Cat# 74004 |
| Ovation RNA-Seq System V2 | NuGEN/Tecan | Cat# 7102 |
| Illumina Stranded Total RNA Prep with Ribo-Zero Plus | Illumina | Cat# 20040529 |
| NEBNext Library Quant Kit for Illumina | New England Biolabs | Cat# E7630S |
| **Software** | | |
| ImageJ | NIH | |
| Imaris | Bitplane | |
| GraphPad Prism | https://www.graphpad.com/ | |
| Seurat | Satija Lab | |

| Reagent/resource | Reference or source | Identifier or catalog number |
|---|---|---|
| snakePipes | GitHub | |
| **Other** | | |
| Leica CM1900 Cryostat | Leica Microsystems | |
| Leica TCS SP8 Confocal Microscope | Leica Microsystems | |
| Zeiss LSM 780 Confocal Microscope | Zeiss | |
| FACSAria III Cell Sorter | BD Bioscience | |
| Incucyte S3 Live-cell Analysis System | Sartorius | |

## Mouse breeding and husbandry

Mouse breeding and husbandry were performed and monitored at the Max Planck Institute of Immunobiology and Epigenetics in Freiburg and the University Hospital Rechts der Isar of the Technical University (MRI-TUM) in Munich following the guidelines set provided by the Federation of European Laboratory Animal Science Association and as approved by German authorities. The mice were kept in groups of three or four in Greenline IVC GM500 plastic cages. They were housed in a temperature-controlled environment ($21 \pm 2\,°C$) on a 12 h light/dark cycle and given ad libitum access to food and water in the animal facilities. All mouse strains in this study were without a health burden. Mice were only used for organ removal after euthanasia by carbon dioxide exposure or cervical dislocation under isoflurane anesthesia, and thus not subject to experimental procedures and ethical approval according to §4 (3) Tierschutzgesetz. The following mouse lines were used in the study: Wild-type C57BL/6J mice were from Charles River. $Arpc4^{flox}$ mice on the C57BL/6J background were provided by Prof. Metello Innocenti at the Università degli Studi di Milano-Bicocca (van der Kammen et al, 2017). $Tg(Cx3cr1\text{-}Cre)$ MW126Gsat/Mmucd were generated by Nathaniel Heintz, Ph.D., The Rockefeller University, GENSAT, and purchased from MMRRC (stock #036395-UCD) (Crotti et al, 2014). $Cx3cr1^{GFP}$ knock-in mice were from Jackson Laboratory (Strain No.: 005582). Since our initial characterization of the control genotypes $Tg(Cx3cr1\text{-}Cre)$ $Arpc4^{+/+}$ and Cre-negative $Arpc4^{fl/fl}$ mice did not show any differences in microglia parameters relevant for this study (Figs. EV2 and EV3), we decided to use Cre-negative $Arpc4^{fl/fl}$ mice as control animals for comparisons with age-matched littermate $Arpc4^{\Delta Cx3cr1}$ mice, allowing us to reduce surplus mouse generation in animal breedings, according to 3R ethical guidance principles. For the majority of experiments, adult $Tg(Cx3cr1\text{-}Cre)$ $Arpc4^{fl/fl}$ ($Arpc4^{\Delta Cx3cr1}$) mice were compared with age-matched littermate $Arpc4^{fl/fl}$ control mice. For two-photon experiments, adult $Tg(Cx3cr1\text{-}Cre)$ $Arpc4^{fl/fl}$ $Cx3cr1^{GFP/+}$ mice were compared with age-matched littermate $Arpc4^{fl/fl}$ $Cx3cr1^{GFP/+}$ control mice. $Tg(Cx3cr1\text{-}Cre)$ was used heterozygous in all experiments.

Mice were bred in-house, and pups were kept with the adult female under standard light/dark conditions until 3 weeks of age, before they were weaned. Most of the experiments, including immunohistochemistry, RNAscope, and RNA sequencing, were performed on adult mice at 3 months of age. Both males and females were included in all analyses; we did not measure any influence of sex on our analyzed parameters in the study (Fig. EV4B). For microglia isolation, P6–P8 C57BL/6 J wildtype or knockout mice were used. The mice and samples, including brain sections, were allocated into experimental groups randomly.

## Mouse perfusion and immunohistochemistry

- Euthanize mice with $CO_2$.
- Perfuse immediately:
  - Make a small incision in the right atrium.
  - Start HBSS saline perfusion using a 20 mL dosing syringe.
- Dissect brains and postfix in 4% paraformaldehyde (PFA) overnight.
- Cryoprotect brains in 30% sucrose in PBS for ≥1 day.
- Embed/freeze on dry ice using Tissue-Tek OCT.
- Cut 30 µm coronal sections using a cryostat (Leica CM1900).
- Collect free-floating sections into 25% glycerol + 25% ethylene glycol in PBS.
- Rinse sections in 1× PBS + 0.2% Tween-20.
- Permeabilize in 0.5% Triton X-100 for 10–30 min (depending on primary antibody).
- Block endogenous mouse IgG:
  - Incubate with Fab fragment goat anti-mouse IgG (1:100; Dianova) for 1 h at RT.
- Block sections in 2.5% FCS + 2.5% BSA + 2.5% fish gelatin in PBS for 1 h at RT.
- Incubate with primary antibodies (diluted in 10% blocking solution) overnight at 4 °C.
- Incubate with secondary antibodies (in 10% blocking solution) for 1 h at RT.
- Wash with PBS, then distilled $H_2O$.
- Mount on Superfrost Plus slides using fluorescence mounting medium.
- For AXL staining, additionally perform antigen retrieval using citrate buffer (10 mM, pH 6) on free-floating sections.

## DAB immunohistochemistry staining

- Use VECTASTAIN® ABC Kit (standard; Vector Laboratories).
- Transfer free-floating 30 µm sections into a 24-well plate.
- Wash sections 3× in 1× PBS.
- Block endogenous peroxidase:
  - Incubate in 3% $H_2O_2$ for 20 min at 4 °C.
  - Wash with PBS.
- Incubate in 100% blocking solution (2.5% FCS + 2.5% BSA + 2.5% fish gelatin in PBS) for 20 min at RT.
- Incubate with primary antibodies (in 10% blocking solution) overnight at 4 °C.
- Wash with PBS.
- Incubate with biotinylated secondary antibodies for 1 h at RT.
- Wash with PBS.
- Prepare ABC solution:
  - Add 20 µL reagent A (Avidin DH) + 20 µL reagent B (biotinylated HRP H) in PBS.
  - Incubate sections in ABC solution for 30 min at RT.
- Wash 3×.

- Develop signal using DAB staining reagents.
- Stop reaction with distilled water.
- Wash with PBS and place sections on Superfrost Plus slides to dry.
- Rehydrate by decreasing ethanol concentrations in water.
- Counterstain with hematoxylin for 1 min.
- Dehydrate by increasing alcohols.
- Clear in xylol (Chemie Vertrieb GmbH).
- Mount with Depex.

## RNAscope in situ hybridization

- Use RNAscope Multiplex Fluorescent Detection Reagents v2 (ACD).
- Fix 30 µm cryosections on Superfrost Plus slides at −20 °C for 3 days.
- Pretreat sections:
  ○ Hydrogen peroxide for 10 min at RT.
  ○ Antigen retrieval by boiling for 12 min to unmask the target RNA.
- Apply Protease III for 30 min at 40 °C.
- Hybridize probe:
  ○ Incubate with mouse Ms4a7 probe (channel 1) for 2 h at 40 °C.
- Run controls:
  ○ Positive controls: Polr2a, Ppib, Ubc
  ○ Negative control: DapB
- Perform amplification and detection per kit instructions.
- Detect signal using TSA Vivid Fluorophore 570 diluted 1:2500 in TSA buffer.
- After ISH, perform IHC for microglia using IBA1 (Wako, 1:1000).
- Counterstain nuclei with DAPI.
- Mount with ProLong Gold Antifade (Invitrogen).

For more details, see https://static.yanyin.tech/literature_test/protocol_io_true/protocols.io.e6nvwk5p2vmk/i7wkbpw7x3.pdf.

## Confocal microscopy of microglia and image analysis

To analyze microglia numbers and morphology, confocal stacks (step size: 0.8 µm) were captured in the *z*-direction from indicated brain regions with 20× or 40× objectives of a Leica TCS SP8 and Zeiss LSM 780 confocal fluorescence microscopes. The DAB images were acquired via ZEISS Axio Imager A2. Images were processed and analyzed with the image processing software, including Imaris (version 9.6.1, Bitplane, Zurich, Switzerland) and ImageJ (version 1.53, National Institutes of Health, Bethesda, MD, USA; Schneider et al, 2012). A three-dimensional image was generated in the Imaris Surpass view. The cell density was determined using the spot function in Imaris software manually. To perform unbiased and automatic analysis of microglial morphometric parameters, including ramification and cell area, we used a MATLAB-based program, 3DMorph, relying on automatic, signal-threshold-based reconstruction, which is publicly available on Github (https://github.com/ElisaYork/3DMorph) (York et al, 2018). For the reconstruction of microglia cells, 30µm coronal brain sections were imaged with a 40x objective of the Zeiss LSM 780 confocal microscope. All images were processed with the following settings: Otsu thresholding 0.5–1.5; noise filtration 200–500. 30 to 40 cells were analyzed. The largest and smallest cells were defined on a per-image basis to maximize

cell yield. Lastly, after reconstruction, the cells analyzed were quality-controlled in 3D to make sure that the cells segmented in one stack are cells that are not fused or cut off in the confocal stack. To ensure an unbiased result, the investigator was blinded. Cells localized on image borders in the X, Y, or Z plane were excluded from analysis. All movies were created using the open-source software ImageJ.

## Live-cell imaging of microglia in brain slices

### Preparation of mouse brain slices

Mice were euthanized by decapitation following anesthesia with isoflurane. Acute 300-µm-thick brain slices were obtained from the hippocampus using a Leica VT1200 S microtome under protective conditions (Kyrargyri et al, 2019; Ting et al, 2014). Slicing was performed in 4 °C cold cutting solution containing: 87 mM NaCl, 25 mM NaHCO$_3$, 2.5 mM KCl, 0.5 mM CaCl$_2$, 3 mM MgCl$_2$, 1.25 mM NaH$_2$PO$_4$, 25 mM glucose, 75 mM sucrose (pH 7.4, 315 mOsm/l). The slices were then allowed to recover at 34–36 °C in the cutting solution for 12–15 min before being transferred to a bicarbonate-buffered artificial cerebrospinal fluid (aCSF) at room temperature. This solution contained 125 mM NaCl, 2.5 mM KCl, 25 mM NaHCO$_3$, 1.25 mM NaH$_2$PO$_4$, 2 mM CaCl$_2$, 1 mM MgCl$_2$, and 10 mM Glucose. Both solutions were continuously oxygenated with a 95% O$_2$/5% CO$_2$ gas mixture. Experiments were performed within 4 h of slicing to maintain microglial cells closely resembling the in vivo state, thereby reducing activation (Gyoneva and Traynelis, 2013; Kyrargyri et al, 2019).

### Two-photon imaging

Microglia were imaged in acute brain slices at a depth of 50–150 µm beneath the slice surface using a Nikon A1R+ multiphoton microscope equipped with a 25X lens (NA 1.1) and a Spectra-physics laser (Mai Tai Insight Deepsee Dual), with brain slices continuously perfused with aCSF during imaging. For time-lapse imaging of microglial surveillance, 50 µm stacks with 2 µm z-intervals were acquired every minute using the laser tuned to a wavelength of 920 nm at a pixel dwell time of 2.2 µs. Images typically had a resolution of 512 × 512 (for live imaging), covering a square field of view of 258 µm. To study directed process motility (chemotaxis), a small tissue volume (10 µm) was ablated by briefly increasing laser power to 95% for 3–5 s. Image stacks of 25 µm thickness at 2 µm z-intervals were acquired every 30 s, with the laser ablation site centered in the field of view.

### Analysis of imaging data

The imaging data for microglial surveillance and chemotaxis were processed using ImageJ and MATLAB (Version R2024A, MathWorks), following established methods (Kyrargyri et al, 2019; Madry et al, 2018) and custom-written code available at https://github.com/AttwellLab/Microglia. Key steps included background subtraction, median filtering, and correction for lateral and z-drift in 4D hyperstacks. A combination of automated and manual image processing steps was used. Only microglial cells that remained fully contained within the acquired z-stack throughout the entire imaging period were included in the analysis. This manual cell selection step was used as a quality-assurance measure to avoid partial-volume artifacts or incomplete cell reconstructions. Individual microglia were then manually thresholded to generate binary masks, as peripheral processes frequently exhibit lower fluorescence intensity compared to somata, rendering automated thresholding

unreliable. All imaging analyses were performed with the experimenter blinded to the experimental conditions/genotypes. For surveillance analysis, the addition or removal of pixels (representing process extensions, PE, and process retractions, PR) was calculated frame by frame. The resulting "surveillance index" was defined as the sum of non-zero pixels in both PE and PR and corresponds to the brain volume surveilled by the cell over time, reflecting both the rate and spatial extent of process dynamics. To additionally assess morphological complexity independent of process motility, a ramification index was calculated as the ratio of the cell perimeter to its area, normalized to the perimeter of a circle with the same area. Together, these indices enable discrimination between reduced surveillance caused by loss of process complexity versus reduced movement of otherwise intact processes. For chemotaxis analysis, both manual and automated image processing steps were used. First, maximum-intensity projections of the 4D image stacks were generated, and microglial processes were manually thresholded to create binary masks, for the same reasons described for the surveillance analysis (i.e., heterogeneous fluorescence intensities across soma and peripheral processes). Next, the center of the laser ablation was manually defined by placing a hairpin marker at the ablation site. Based on this manually defined center, the software then automatically generated 36 radial sectors around the ablation site. At each time point, the distance between the leading edge of the chemotactic microglial processes and the defined ablation center was quantified, representing the progression of directed process extension toward the lesion over time. As chemotaxis proceeded, this distance progressively declined until processes converged at the ablation site. Each chemotactic response was treated as one biological unit, reflecting the coordinated response of all microglia involved in the chemotactic event. To compare chemotactic dynamics across animals, the full time-course of distance measurements was summarized by calculating the area under the curve (AUC). Lower AUC values reflect faster chemotactic convergence. AUC values were then averaged per mouse and used for statistical comparison between genotypes.

## Microglial cell isolation

P6–P8 microglia isolation (for culture)

- Dissect brains from P6–P8 mice.
- Dissociate tissue using Neural Tissue Dissociation Kit (Papain; Miltenyi).
- Mechanically dissociate using gentleMACS Octo Dissociator with Heaters (Miltenyi).
- Pass suspension through a 40 µm strainer.
- Wash twice with HBSS and FACS buffer (0.5% BSA + 2 mM EDTA in PBS).
- Sort microglia on FACSAria III (BD):
  - Stain with Zombie Violet viability dye (1:1000).
  - Block with mouse FcR-blocking reagent (CD16/CD32; eBioscience).
  - Label with lineage exclusion antibodies (CD3, CD19, Ly6C, Ly6G) and microglia markers (CD11b, CD45).
  - Sort viable single immune cells:
- Control: CD45$^{low}$ CD11b$^{high}$
- KO: CD45$^{low}$ CD11b$^{low}$/CD11b$^{high}$
- Collect sorted cells into culture medium.
  Adult microglia isolation (for RNA-seq)
- Perfuse adult mice.
- Dissect and homogenize the brain with a Dounce homogenizer.

- Pass through a 70-µm strainer.
- Wash twice in HBSS containing 45% glucose and HEPES.
- Remove myelin by density gradient:
  - Resuspend pellet in 37% Percoll (Sigma) in 1× PBS
  - Centrifuge 800×$g$, 30 min, low acceleration, no brake
  - Remove the top membrane/myelin fraction by vacuum
- Wash pellet twice with FACS buffer (400×$g$, 5 min).
- Stain and gate as above; sort (Control: CD45$^{low}$ CD11b$^{high}$ and KO: CD45$^{low}$ CD11b$^{low}$/CD11b$^{high}$) into lysis buffer for RNA extraction.

## Matrigel 3D culture

- Seed FACS-sorted microglia into 40% Matrigel (Corning) supplemented with:
  - 20% L929 conditioned medium
  - 1.5 µg cholesterol
  - 100 ng/mL IL-34
  - 2 ng/mL TGF
- Prepare ice-cold 96-well Incucyte ImageLock plates.
- Add Matrigel mix containing 8000 cells/well.
- Centrifuge 3 min at 75×$g$, 4 °C to place cells into one focal plane.
- Polymerize gel:
  - Incubate at 37 °C for 30 min
- Rest at RT 10 min, then add 200 µL medium on top.
- Acquire time-lapse imaging using Incucyte S3:
  - Image every 15 min for 18 h
  - Use the ImageLock module and 20× objective

## Intracellular flow cytometry

- Homogenize adult brain tissue with a Dounce homogenizer in HBSS buffer containing:
  1.5% HEPES
  1% of 45% glucose in HBSS
- Pass through a 70-µm strainer.
- Wash once; resuspend pellet in 37% Percoll in PBS 1×.
- Centrifuge 800×$g$ for 30 min, no brake.
- Remove the myelin layer; wash pellet and resuspend in FACS buffer (0.5% BSA + 2 mM EDTA in PBS).
- Stain with Zombie Violet (1:1000) + surface markers (lineage exclusion antibodies, CD3, CD19, Ly6C, Ly6G, and microglia markers,CD11b and CD45).
- Fix/permeabilize with BD Transcription Factor Buffer Set Fix/Perm for 2 h at RT.
- Wash; incubate in Perm/Wash buffer (on ice, 1 h) with:
  unconjugated anti-ARPC4 (1:300)
  anti-ARPC2–Alexa Fluor 488 (1:250)
- Wash twice; incubate 1 h with Alexa Fluor 488–anti-goat secondary (Jackson ImmunoResearch) to detect ARPC2 and ARPC4.
- Wash twice; resuspend in FACS buffer and analyse by flow cytometry.

## Bulk RNA sequencing

The isolated microglia were homogenized in RLT buffer using QIAShredder (QIAGEN), and the total RNA was extracted using

RNeasy Micro Kit (QIAGEN). cDNA was synthesized using Ovation RNA-Seq System V2 (NuGEN). Library Preparation was done according to instructions from Illumina Stranded Total RNA Prep with Ribo-Zero Plus (20040529, Illumina), samples were sequenced paired end (2 × 100 bp) on a NovaSeq6000 (Illumina). Bulk RNA sequencing data were processed using snakePipes version 2.5.1, a comprehensive bioinformatics suite of pipelines built using snakemake and Python (Bhardwaj et al, 2019). Raw reads were quality-checked using FastQC, and adapters were trimmed using Cutadapt. The cleaned reads were aligned to the mouse reference genome (GRCm38) using the STAR aligner. FeatureCounts was employed to quantify gene expression levels. Differential expression analysis was conducted using DESeq2, integrated within the mRNAseq workflow in snakePipes. Genes with a fold change greater than two and *p* values less than 0.05 were considered for further hypergeometric pathway enrichment analysis.

## ScRNA sequencing

For single-cell library preparation, microglia were isolated from four male control, three male *Arpc4*$^{ΔCx3cr1}$, and unintendedly one female *Arpc4*$^{ΔCx3cr1}$ littermate mice, all age-matched and housed under identical conditions. Animal dissection, fluorescence-activated cell sorting (FACS)-based microglia isolation, GEM generation, and cDNA amplification were performed on the same day by the same operator using identical reagents, instrument settings, and a single lot of consumables to minimize technical variation. Viable (DAPI$^-$) single CD45$^+$ cells were sorted from control and *Arpc4*$^{ΔCx3cr1}$ for the analysis. Cells from each mouse were individually barcoded using 10X Feature Barcoding with Cell Multiplex Oligos (CMOs) and counted before pooling equal cell numbers from each animal into a single suspension, from which one pooled library was prepared. This design ensured equal representation of each animal and avoided multiple experimental batches. The female *Arpc4*$^{ΔCx3cr1}$ mouse was excluded from downstream analysis to avoid potential sex-related bias. Cells were loaded onto a Chromium Single Cell 3′ G Chip (10X Genomics) according to the manufacturer's instructions without modifications using Chromium Next GEM Single Cell 3' Reagents Kit v3.1 (10X Genomics). Individual biological samples were multiplexed using Cell Multiplex Oligos (CMOs) for CellPlex reagents. Dual Index Plate NN Set A (10x Genomics) was used for Cell Multiplexing library construction, and Dual Index Plate TT Set A was used for Gene Expression library construction according to the manufacturer's instructions. The cDNA content and size of post-sample index PCR samples was analyzed using a 2100 BioAnalyzer (Agilent). Library quantification was done using NEB Next® Library Quant Kit for Illumina® (New England Biolabs) following manufacturer's instructions. Sequencing libraries were loaded on an Illumina NextSeq 1000 P2 flow cell, with sequencing settings according to the recommendations of 10× Genomics (read 1: 28 cycles, read 2: 90 cycles, i7 index: 10 cycles, i5 index: 10 cycles). Cell Ranger v7.1.0 software was used to align the sequencing data to the mouse genome mm10. The indexed mouse genome used for alignment was downloaded from 10x genomics (refdata-gex-mm10-2020-A). The single-cell data were loaded into R and into a Seurat object (v.4) for demultiplexing and subsequent bioinformatics analysis. Cells with a high mitochondrial content (>20%) or a low number of genes detected (<200) were discarded. SoupX (Young and Behjati, 2020) was used to estimate and remove cells contaminated by ambient mRNA. The data were normalized with the "LogNormalized" method and scaled. The top 10,000 variable genes were used for the principal component analysis, and the UMAP was generated using the top 20 principal components. The Louvain algorithm with resolution = 0.8 from the FindClusters function was used to assign clusters to cells in an unbiased way. Cell type annotation was performed manually using common markers as reference. Other bioinformatics analyses and data visualization were performed with the other Seurat or ggplot2 functions.

## Quantification and statistical analysis

One-way analysis of variance (ANOVA) followed by the Bonferroni post hoc test was applied to compare more than two groups. A two-tailed Student's *t*-test was performed for comparison of two groups. Two-way ANOVA followed by a Bonferroni post hoc test was used for analyzing the interaction of different variables (age, genotype, brain region, or treatment reagent). For both surveillance and chemotaxis analyses, data were first aggregated at the level of the individual animal. Surveillance and ramification indices were averaged per cell and then across all analyzed cells per animal. For chemotaxis, AUC values from individual chemotactic events were averaged per mouse. These per-mouse mean values served as the statistical units for group comparison, ensuring that each animal contributed a single data point. Normality and equality of variances were assessed prior to testing. Depending on test assumptions, either a two-sided Student's *t*-test (equal variances) or a Welch's *t*-test (unequal variances) was applied. Statistical analyses were done using GraphPad Prism (GraphPad Software, Inc.). A *P* value of <0.05 was considered significant. The type of statistical test and the *P* value for each experiment are included in the figure legends. All cell culture experiments using primary cells were done at least three times independently to ensure reproducibility. Technical and biological replicates were included. The average of at least three technical replicates was counted as one biological replicate, which was then used for statistical analysis and comparison within the biological replicates. For all mouse experiments, three to four mice per genotype were analyzed. To account for variability within the biological sample for histological analysis, three random brain sections per animal were quantified, and three to four random regions of interest per brain section were analyzed. All values obtained from in vivo and in vitro experiments were represented as mean ± s.d. Sample sizes were chosen based on prior literature using similar experimental paradigms. The calculated Cohen's *d*, showing the effect size in each analysis in this work, can be found in the statistics table in Dataset EV1. The value of *n* per group and what n represents in each experiment can be found in the figure legends. Data acquisition and analysis for wild-type and knockout samples were done in a blinded manner, and no data were excluded from any analysis. Only in the scRNAseq data (see before), data from one unintendedly sequenced female *Arpc4*$^{ΔCx3cr1}$ mouse was excluded from downstream analysis to avoid potential sex-related bias.

# Data availability

The single-cell RNA sequencing data generated during this study are available in the NCBI database through accession number

GSE288625. The data for Bulk RNA sequencing have been deposited in the European Nucleotide Archive (ENA) at EMBL-EBI under accession number PRJEB101196. All other data that support the findings are available as a source data file.

The source data of this paper are collected in the following database record: biostudies:S-SCDT-10_1038-S44319-026-00721-8.

## Peer review information

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

## Acknowledgements

This project was initiated at the Max Planck Institute of Immunobiology and Epigenetics in Freiburg and completed at TUM Klinikum rechts der Isar in Munich. It was supported by grants from the German Research Foundation (CRC/TRR167, Project ID 259373024 and Lump Sum Funding to SS). TL was further supported by the Max Planck Society and project grants from SFB1450 (Project ID 431460824), TRR332 (Project ID 449437943), and SFB1348

(Project ID 386797833). This work was also supported by grants from the BMBF/DLR (01EE2303B to JP). For the Bulk RNA sequencing, we thank the Deep sequencing and Bioinformatics facilities, and for cell sorting and microscopic imaging, we are grateful for the support from the Core facilities at the Max Planck Institute of Immunobiology and Epigenetics. We thank the Lighthouse Core Facility, University of Freiburg, for their support with FACS-based cell sorting. The Lighthouse Core Facility is funded in part by the Medical Faculty, University of Freiburg (Project Numbers 2023/A2-Fol; 2021/B3-Fol), the DKTK, and the DFG (Project Number 450392965). We thank Thomas Misgeld at the Technical University of Munich (TUM) for giving us access to confocal microscopy in Munich. Roie Dvir was supported by the Elite Network of Bavaria (Elite Graduate Program Biomedical Neuroscience, S-LW-2016-351/2). Katrin Kierdorf was supported by the Heisenberg program of the DFG (Project ID 544402801), the DFG under Germany's Excellence Strategy (grant no. CIBSS—EXC-2189, Project ID 390939984) and project grants within the TRR359 (Project ID 491676693), the CRC1479 (Project ID 441891347), CRC1160 (Project ID 256073931), and the TRR167 (Project ID 259373024).

## Author contributions

**Shima Safaiyan**: Conceptualization; Data curation; Formal analysis; Funding acquisition; Validation; Investigation; Visualization; Methodology; Writing—original draft; Project administration; Writing—review and editing; performed and analyzed experiments of Figs. 1D, 3A,B,G,H,I, 4A, 5F,G, and EV1A–D,F, EV2A–C, EV3B–E. **Maximilian Frosch**: Resources; Data curation; Formal analysis; Validation; Investigation; Methodology; Writing—review and editing; performed and analyzed experiments of Figs. 1A,C, 3C–F, 5A–E, 6A–C and EV3A, EV4A–D, EV5. **Tom Bickel**: Resources; Data curation; Formal analysis; Validation; Investigation; Visualization; Methodology; Writing—review and editing; performed and analyzed experiments of Fig. 2A-D, Fig. 4B-C. **Gianni Monaco**: Formal analysis; Investigation; Methodology; analyzed experiments of Figs. 5A–E and 6A–C. **Roie Dvir**: Formal analysis; Investigation; Methodology; performed and analyzed experiments of Figs. 5F,H and 6D. **Christian Madry**: Resources; Formal analysis; Supervision; Funding acquisition; Validation; Investigation; Methodology; Writing—review and editing. **Lance Fredrick Pahutan Bosch**: Formal analysis; Investigation; Methodology; analyzed experiments of Figs. 1B and EV1E and EV3B. **Katrin Kierdorf**: Software; Supervision; Funding acquisition; Validation; Visualization; Methodology; Writing—review and editing. **Metello Innocenti**: Resources; Writing—review and editing. **Josef Priller**: Resources; Funding acquisition; Writing—review and editing. **Marco Prinz**: Resources; Funding acquisition; Methodology; Writing—review and editing. **Tim Lämmermann**: Conceptualization; Supervision; Funding acquisition; Visualization; Methodology; Writing—original draft; Project administration; Writing—review and editing.

Source data underlying figure panels in this paper may have individual authorship assigned. Where available, figure panel/source data authorship is listed in the following database record: biostudies:S-SCDT-10_1038-S44319-026-00721-8.

## Funding

## Disclosure and competing interests statement

The authors declare no competing interests.

# Expanded View Figures

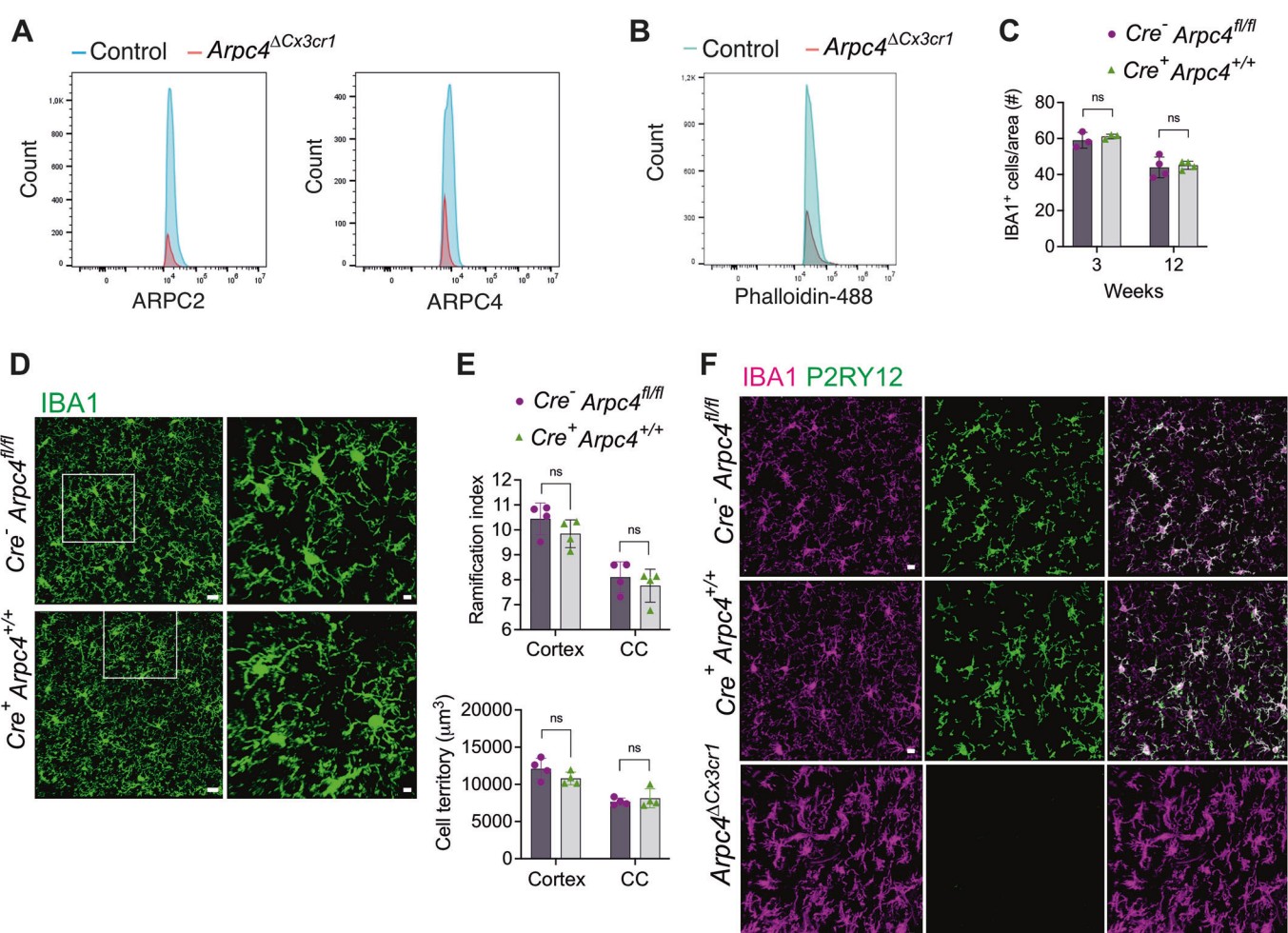

**Figure EV1. Microglia characteristics in *Tg(Cx3cr1-Cre)*-expressing and *Tg(Cx3cr1-Cre)*-negative *Arpc4*⁺/⁺ mice.**

(A) Flow cytometric analysis of ARPC2 and ARPC4 expression in 12-week-old *Tg(Cx3cr1-Cre) Arpc4*^fl/fl^ (*Arpc4*^ΔCx3cr1^) and *Arpc4*^fl/fl^ control mice (Biological replicates: $n = 4$ mice per group, Technical replicates: $n = 3$ experiments). (B) Flow cytometric analysis of Alexa Fluor 488-conjugated phalloidin, indicating global levels of F-actin in microglia from 12-week-old *Arpc4*^ΔCx3cr1^ and control mice (Biological replicates: $n = 4$ mice per group, Technical replicates: $n = 2$ experiments). (C) Quantification of microglial cell number in the cortex of 3- and 12-week-old *Cre*⁺ *Arpc4*⁺/⁺ and *Cre*⁻ *Arpc4*^fl/fl^ mice ($n = 3$-4 mice per group, data were presented as mean ± s.d, two-tailed Student's *t*-test, ns: $P > 0.9999$). (D, E) Representative images of IBA1⁺ microglia morphology and quantification of microglia ramification and cell territory in 12-week-old *Tg(Cx3cr1-Cre) Arpc4*⁺/⁺ (*Cre*⁺ *Arpc4*⁺/⁺) and Cre-negative *Arpc4*^fl/fl^ (*Cre*⁻ *Arpc4*^fl/fl^) mice. Scale bar: 20 μm (left) and 5 μm (right) ($n = 4$ mice per group, data are presented as mean ± s.d., two-tailed Student's *t*-test, ns not significant). (F) Representative images showing IBA1⁺/P2RY12⁺ microglia in 12-weeks-old *Cre*⁺ *Arpc4*⁺/⁺, *Cre*⁻ *Arpc4*^fl/fl^ and *Tg(Cx3cr1-Cre) Arpc4*^fl/fl^ (*Arpc4*^ΔCx3cr1^) mice, scale bar: 20 μm. Source data are available online for this figure.

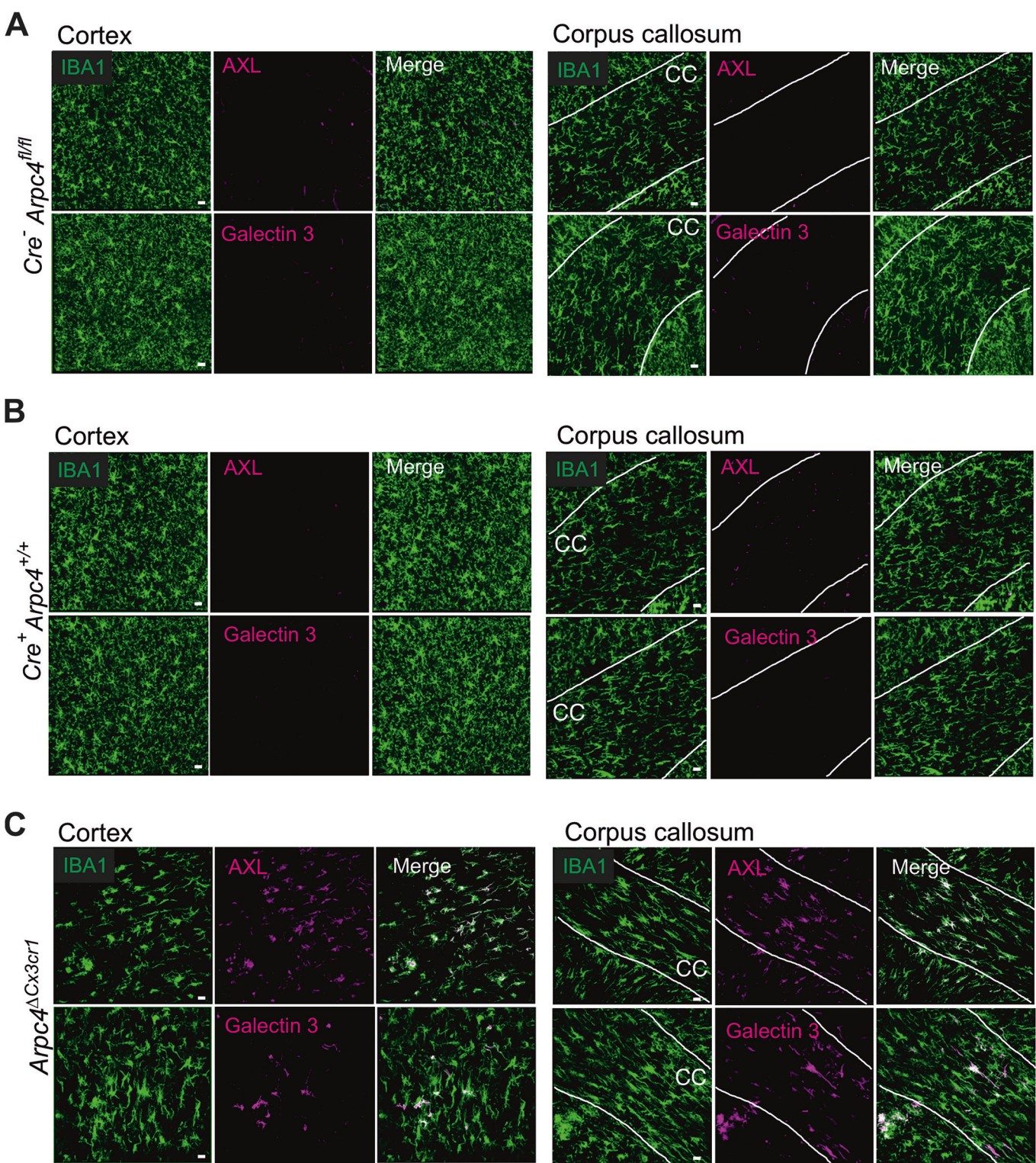

**Figure EV2. Microglial activation in Cre⁺ Arpc4⁺/⁺ compared to the control (Cre⁻ Arpc4ᶠˡ/ᶠˡ) and Arpc4ᐃCx3cr1 mice.**

Representative images of AXL and Galectin-3 expression in IBA1⁺ microglia in the cortex and corpus callosum in 12-weeks-old *Cre-negative Arpc4ᶠˡ/ᶠˡ (Cre⁻ Arpc4ᶠˡ/ᶠˡ)* (**A**), *Tg(Cx3cr1-Cre) Arpc4⁺/⁺ (Cre⁺ Arpc4⁺/⁺)* (**B**), *and Tg(Cx3cr1-Cre) Arpc4⁺/⁺ (Arpc4ᐃCx3cr1)* (**C**) mice. Scale bar: 20 μm. Confocal microscopy overview images come from stitched tile scans of immunostained brain tissues. Source data are available online for this figure.

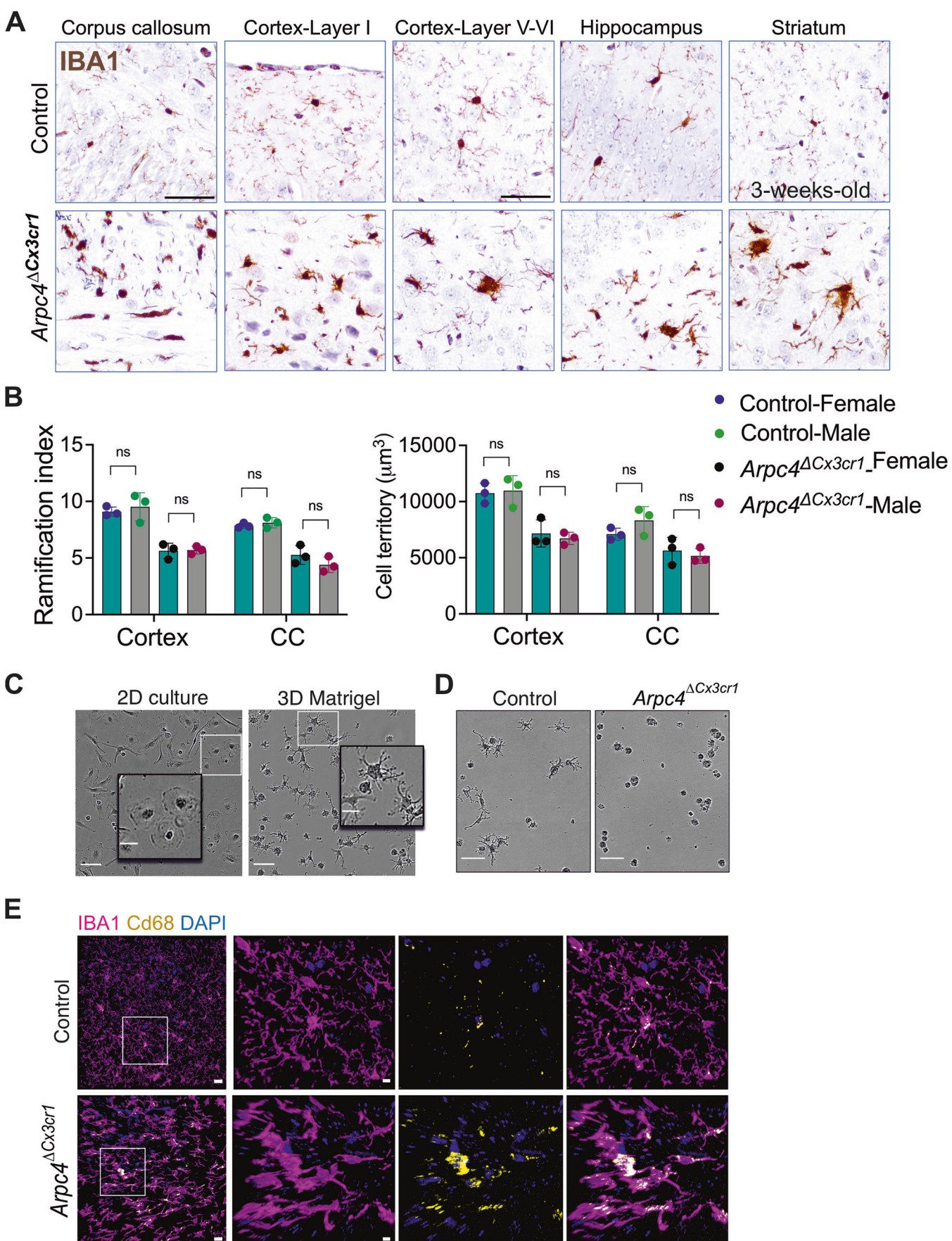

**Figure EV3. Microglial morphology upon *Arpc4* depletion in vivo and in vitro.**

(**A**) Representative DAB-stained images of microglial morphology in different brain regions in 3-week-old *Arpc4*<sup>ΔCx3cr1</sup> and control mice. Scale bar: 100 μm. (**B**) Quantification of microglia ramification and cell territory in 12-week-old female and male *Arpc4*<sup>fl/fl</sup> (control) and *Tg(Cx3cr1-Cre) Arpc4*<sup>fl/fl</sup> (*Arpc4*<sup>ΔCx3cr1</sup>) mice ($n = 3$ mice per group, data are presented as mean ± s.d., two-tailed Student's *t*-test, ns: $P > 0.9999$). (**C**) Comparison of wild-type microglia morphology in 2D and Matrigel 3D culture systems. Scale bar: 50 μm. (**D**) *Arpc4*<sup>ΔCx3cr1</sup> and control microglia in the Matrigel 3D culture after 10 h. Scale bar: 50 μm. (**E**) Representative images of CD68 immunolabeling of IBA1<sup>+</sup> microglia in 3-week-old *Arpc4*<sup>ΔCx3cr1</sup> and control mice. Scale bar: 20 μm (overview), 7 μm (zoom-in). Source data are available online for this figure.

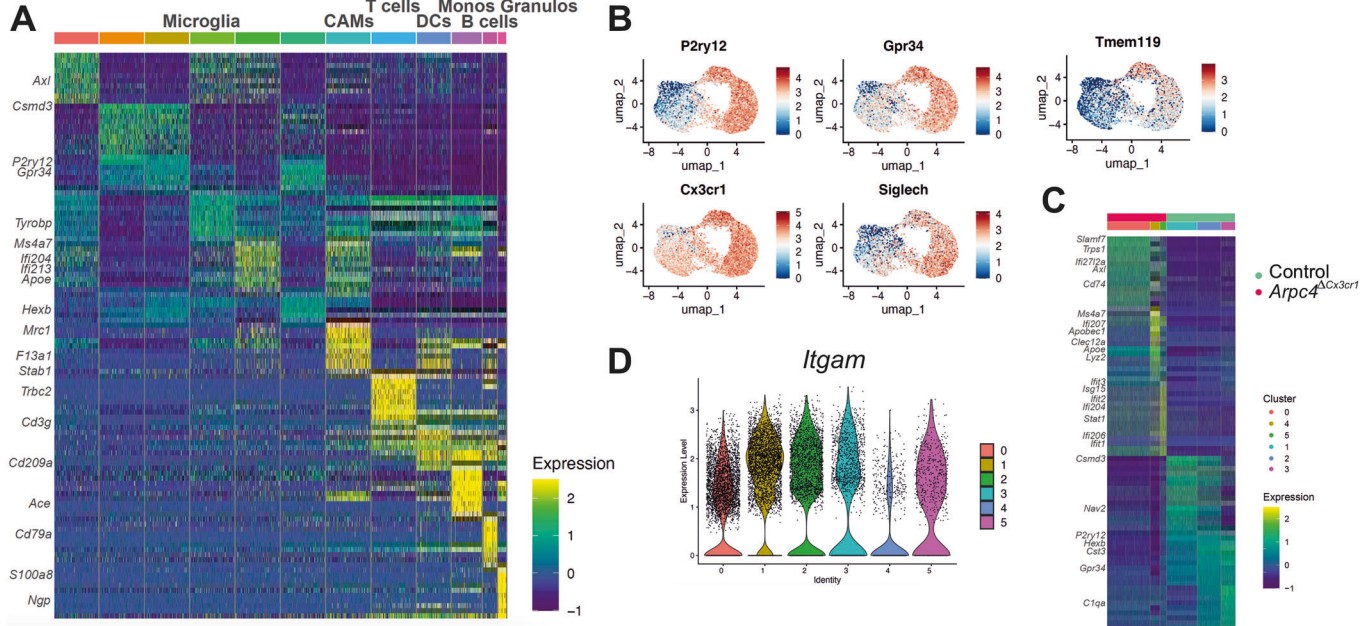

**Figure EV4. Downregulation of TGFβ-dependent microglial homeostatic signature, their reactive phenotype in Arpc4^ΔCx3cr1 mice.**

(A) Heat map of genes (rows) of all the cell types that are highlighted in Fig. 3A, B. Key genes are highlighted. Colors in the heat map correspond to normalized, scaled expression. (B) Expression levels of microglial homeostatic genes, *P2ry12, Gpr34, Cx3cr1, Siglech,* and *Tmem119* in microglia clusters. (C) Heat map of genes (rows) of microglia clusters that are highlighted in Fig. 3C. Key genes are highlighted. Colors in the heat map correspond to normalized, scaled expression. (D) Violin plots showing the expression level of *Itgam* in the microglial clusters in *Arpc4^ΔCx3cr1* and control mice (*n* = 4 control and 3 *Arpc4^ΔCx3cr1* mice).

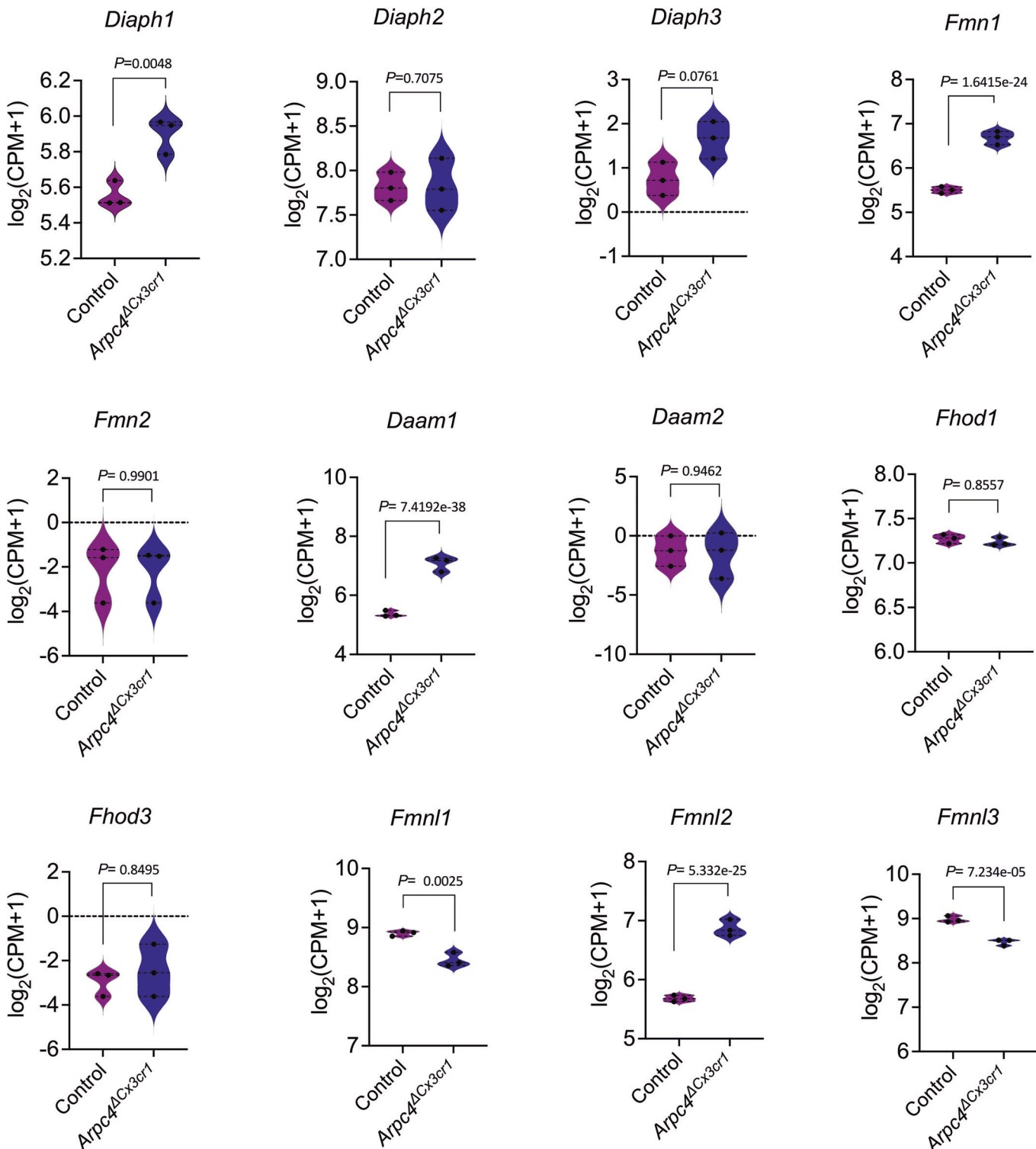

**Figure EV5.  Expression levels of formin/diaphanous family of proteins.**

Violin plots showing gene expression levels of formin/diaphanous family of proteins in microglia isolated from $Arpc4^{\Delta Cx3cr1}$ and control mice ($n = 4$ control and 3 $Arpc4^{\Delta Cx3cr1}$ mice). Data were retrieved from bulk RNA-seq data sets from CNS-isolated microglia, analyzed in Fig. 3C–F. CPM counts per million.

