## [Peer Review File · EMBO Reports]

The Arp2/3 complex controls the development of homeostatic microglia

Shima Safaiyan, Maximilian Frosch, Tom Bickel, Gianni Monaco, Roie Dvir, Lance Bosch, Christian Madry, Katrin Kierdorf, Metello Innocenti, Josef Priller, Marco Prinz, and Tim Lämmermann

Corresponding author(s): Tim Lämmermann (laemmermann@uni-muenster.de) , Shima Safaiyan (shima.safaiyan@tum.de)

Review Timeline:

Submission Date:	30th May 25
Editorial Decision:	25th Jul 25
Revision Received:	26th Nov 25
Editorial Decision:	12th Jan 26
Revision Received:	30th Jan 26
Accepted:	2nd Feb 26

Editor: Achim Breiling

Transaction Report:

Dear Dr. Lämmermann,

I have already forwarded to you the reports I received from the three referees that I asked to evaluate your study. Please find them again below. After going through your preliminary point-by-points response (revision plan), I feel that all referee points will be adequately addressed.

Given the constructive referee comments, I would thus like to invite you to revise your manuscript with the understanding that the concerns of the referees must be addressed in the revised manuscript and/or in a detailed point-by-point response, as indicated in your revision plan. Acceptance of your manuscript will depend on a positive outcome of a second round of review. It is EMBO reports policy to allow a single round of revision only and acceptance of the manuscript will therefore depend on the completeness of your responses included in the next, final version of the manuscript.

1) a .docx formatted version of the final manuscript text (including legends for main figures, EV figures and tables), but without the figures included. Figure legends should be compiled at the end of the manuscript text.

2) individual production quality figure files as .eps, .tif, .jpg (one file per figure), of main figures and EV figures. Please upload these as separate, individual files upon re-submission.

4) a complete author checklist, which you can download from our author guidelines

(<https://www.embopress.org/page/journal/14693178/authorguide>). Please insert page numbers in the checklist to indicate where the requested information can be found in the manuscript. The completed author checklist will also be part of the RPF.

5) that primary datasets produced in this study (e.g. RNA-seq, ChIP-seq, structural and array data) are deposited in an appropriate public database. If no primary datasets have been deposited, please also state this in a dedicated section (e.g. 'No primary datasets have been generated and deposited'), see below.

The accession numbers and database should be listed in a formal "Data Availability" section that follows the model below. This is now mandatory (like the COI statement). Please note that the Data Availability Section is restricted to new primary data that are part of this study. This section is mandatory. As indicated above, if no primary datasets have been deposited, please state this in this section

Data availability

6) We now request the publication of original source data with the aim of making primary data more accessible and transparent to the reader. You will receive a separate email with instructions for providing source data with your revised manuscript, including information how to upload and organize the files.

8) Regarding data quantification and statistics, please make sure that the number "n" for how many independent experiments were performed, their nature (biological versus technical replicates), the bars and error bars (e.g. SEM, SD) and the test used to calculate p-values is indicated in the respective figure legends (also for EV and Appendix figures). Please also check that all the p-values are explained in the legend, and that these fit to those shown in the figure. Please provide statistical testing where applicable. Please avoid the phrase 'independent experiment', but clearly state if these were biological or technical replicates. Please also indicate (e.g. with n.s.) if testing was performed, but the differences are not significant. In case n=2, please show the data as separate datapoints without error bars and statistics. See also:
<http://www.embopress.org/page/journal/14693178/authorguide#statisticalanalysis>

9) Please add scale bars of similar style and thickness to all microscopic images, using clearly visible black or white bars (depending on the background). Please place these in the lower right corner of the images themselves. Please do not write on or near the bars in the image but define the size in the respective figure legend.

10) Please also note our reference format:

12) We now use CRediT to specify the contributions of each author in the journal submission system. CRediT replaces the author contribution section. Please use the free text box to provide more detailed descriptions and do NOT provide your final manuscript text file with an author contributions section. See also our guide to authors:
<https://www.embopress.org/page/journal/14693178/authorguide#authorshipguidelines>

13) All Materials and Methods need to be described in the main text using our 'Structured Methods' format, which is required for all research articles. According to this format, the Methods section should include a Reagents and Tools Table (listing key reagents, experimental models, software, and relevant equipment and including their sources and relevant identifiers), uploaded as separate file, and a Methods section in which we encourage the authors to describe their methods using a step-by-step

protocol format with bullet points, to facilitate the adoption of the methodologies across labs. More information on how to adhere to this format as well as downloadable templates (.doc) for the Reagents and Tools Table can be found in our author guidelines (section 'Structured Methods'):

14) Please order the manuscript sections like this, using only these names:

Title page - Abstract - Keywords - Introduction - Results - Discussion - Methods - Data availability section - Acknowledgements (please put here all the finding information) - Disclosure and Competing Interests Statement - References - Figure legends - Expanded View Figure legends

15) Please make sure that all the funding information is also entered into the online submission system and that it is complete and similar to the one in the acknowledgement section of the manuscript text file.

16) Please name the movie files 'Movie EV1', 'Movie EV2' and 'Movie EV3' in all places (source file names, titles in the submission system, their callouts and legends). Please provide each legend as a readme.txt file that then should be ZIPped up with its corresponding movie and uploaded (so that we have one zip folder per movie). Finally, please remove the movie legends from the manuscript main text file.

17) Please define each abbreviation the first time it is used in the text. Then please remove the list of abbreviations from the manuscript text.

Please note that corresponding authors are required to supply an ORCID ID upon submission of a revised manuscript. Please do that for co-corresponding author Shima Safaiyan. Please find instructions on how to link the ORCID ID to the account in our manuscript tracking system in our Author guidelines:

<http://www.embopress.org/page/journal/14693178/authorguide#authorshipguidelines>

I look forward to seeing a revised form of your manuscript when it is ready.

Yours sincerely,

Referee #1:

The present study uncovers an unknown function of the Arp2/3 complex : preserve the homeostasis of tissue-resident microglia within the central nervous system (CNS). By genetically deleting the essential subunit Arpc4, the authors investigated the role of the Arp2/3 complex in microglia-an immune cell type that is challenging to study outside its native environment. Their findings add to the growing body of evidence suggesting that depletion of Arp subunits and disruption of Arp2/3-mediated actin branching lead to distinct cellular phenotypes, which vary depending on the myeloid cell type examined.

In this study, the authors report that microglia protrusion branching is altered, as well as motility and chemotaxis towards a wound. They also observed an increase in uptake of extracellular material, which confirms the previously described antagonism between cell migration and macropinocytosis/phagocytosis. The authors may want to discuss the potential mechanisms underlying this increase (Formin upregulation or others?).

But the most interesting phenotype is probably the loss of their homeostatic gene signature. Indeed, Arpc4KO microglia display a gene signature that rather resembles the one of DAM (disease associated microglia). This raises the interesting question of whether the DAM phenotype typically observed in neurodegenerative diseases could result from an altered branched actin cytoskeleton.

As all studies carried out by the Lammermann group, experiments and data are of great quality and the manuscript is extremely clear and well-written. I think it should be published as such.

Referee #2:

In the manuscript 'Arp2/3 complex controls microglia dynamics and Homeostasis' Safaiyan et al. describe the effects of a microglia/myeloid cell-specific knockout of Arpc4 on microglia morphology, motility and transcriptome. Overall, the manuscript is

clearly written and contains well-designed and executed experiments and the conclusions are supported by multiple lines of evidence. I have a few recommendations for the authors to further improve and clarify their manuscript

- The role of Arp2/3 in TGF beta signaling as depicted in figure 8 could benefit from some additional analysis of the collected data. In this figure the authors demonstrate an overlap in gene expression changes between the Arpc4 KO and a Tgfbr2 KO, as well as changes in SMAD phosphorylation that is dependent on the deletion of Arpc4. In the discussion it is mentioned that the SMAD data provide mechanistic insight. In order to clarify this mechanistic insight, I think it important to address whether the relevant receptors and downstream molecules of TGFb signalling are expressed to the same degree in the KO microglia? Is the regulation at the level of the activation/downstream signaling, or is at the level of the presence of the receptors? (similar to decreased expression of P2RY12 as shown in the manuscript)
- There is an almost complete lack of overlapping clusters of microglia from wt and KO animals. This can be biological, as suggested by the authors, however it could theoretically also be the result of technical variation induced by the sampling and pooling. The methods section on this part is not very extensive making it not possible to evaluate the contribution of each. For example, to me it is not clear how many animals were included to collect the cells, if they were collected in batches or in 1 experiment and how or if they were pooled. I recommend to expand on this in the methods section, but maybe also include a small graphical overview of the setup in the figure as panel a.
- Throughout the paper IBA1 is used as a marker for microglia. As is well known, IBA1 does not only mark microglia, but myeloid cells in general. Given the shape and location of the cells I am convinced that the analysis have been done mainly on microglia, however I feel it would be appropriate to state in the manuscript that this assumption was made, and that the data could also reflect some other myeloid populations.
- Overall the methodology should be expanded on to enable replication of experiments.

Minor points:

- In figure 2, the GFP images are mistakenly labeled as IBA1, while it is CX3CR1 that is tagged by GFP
- In figure 2 A,B the images representing the timeline: to me it is not fully clear what the white and green parts in these images represent. I recommend including this in the figure legend
- Some of the details in the figures are too small to read, eg the labels in fig 3c and 3d
- The IF images (for example in Fig3 and 6) are a bit small, if space allows, I would recommend to increase their size

Referee #3:

This study addresses the important and interesting question of how the actin cytoskeleton influences the homeostatic identity of microglia. The authors have generated a large dataset, including two-photon live imaging and single-cell sequencing. However, despite the technical quality of some data, the manuscript in its current form cannot be supported for publication. The study suffers from several fundamental flaws in experimental design and interpretation that undermine the validity of its major conclusions.

Major Concerns:

- 1. Essential Experimental Controls:** The study design lacks the necessary controls to isolate the effect of Arpc4 deletion. Firstly, a Cx3cr1-Cre mouse line without the floxed Arpc4 allele is an indispensable control to ensure that expression of the Cre recombinase itself does not contribute to the phenotype. In addition, the use of the Cx3cr1-Cre knock-in line means the conditional knockout animals are, by definition, heterozygous for Cx3cr1 (Cx3cr1-Cre/+). It is plausible that even this 50% reduction in Cx3cr1 expression (haploinsufficiency) could be sufficient to disrupt the baseline motility and surveillance capacity of microglia, independent of any Arpc4 deletion. Additional experiment data comparing wild-type mice to Cx3cr1-Cre/+ mice (without the floxed allele) is required to demonstrate that the motility parameters being measured are intact. This is required to conclude that the observed phenotype is specifically due to the loss of Arp2/3.
- 2. Cx3cr1 Deficiency in Functional Assays:** A critical ambiguity in the manuscript's reporting leads to a potential confound in the functional assays. Central claims about impaired microglial motility, surveillance, and chemotaxis (Figures 2 and 4) are based on experiments using Arpc4ΔCx3cr1 Cx3cr1GFP/+ mice. The methods and figure legends lack the precise details of the breeding strategy and final genotypes for all experimental cohorts, leaving room for interpretation. Given standard genetic conventions, the most direct and indeed, unavoidable, interpretation is that the combined use of these two lines would result in the disruption of both endogenous copies of the Cx3cr1 gene. If this interpretation, which appears to be the case based on the provided description, is accurate, then this constitutes a major methodological flaw with profound implications for the study's conclusions. It has been unequivocally established for more than two decades, through extensive research in the field, that the Cx3cr1 gene product is critically essential for several key microglial functions. These include, but are not limited to, microglial chemotaxis, motility, and their proper infiltration into the CNS. These functions are critically dependent on the very actin-related roles that the authors propose to investigate in this study.
- 3. Biological Model:** Even granting concerns #1 and #2 the benefit of the doubt, the current experimental model, utilizing the Cx3cr1-Cre line, presents a fundamental limitation in its capacity to address the proposed mechanisms of microglial function and maintenance. The critical aspect of this model is that the Cx3cr1-Cre line initiates recombination events during the embryonic

stages of development, specifically targeting microglia derived from yolk-sac progenitors. This means that any observed cellular or functional alterations in the microglia within this model are not indicative of a disruption to an already established homeostatic state in the adult brain. Since the genetic manipulation likely impacts the very formation and maturation of microglia, any observed phenotype is, at best, a manifestation of a developmental defect-an "actinopathy of development" in this context.

4. Statistics and Analysis: The manuscript's statistical framework has several critical deficiencies that span from experimental design to data analysis and reporting. In the methods, the authors state, "No power analyses were used to predetermine sample sizes," but justify the choice based on prior literature. This justification is insufficient. The simple fact that an error, such as the use of underpowered cohorts, is propagated through the literature does not refute that it is an intrinsic statistical flaw. An error remains an error, regardless of how commonly it is made. The use of very small animal numbers ($n=3-4$ per group) is therefore not statistically justified and is a major concern.

Furthermore, the authors state that both males and females were pooled because they "did not notice any influence of sex." Given the very well-documented sexual dimorphism of microglia, a study with this low N lacks any statistical power to detect such differences, making this claim unsubstantiated. A brief statement in the methods is not a substitute for transparently reporting the statistical analysis; to properly justify pooling the data, a two-way ANOVA (genotype x sex) should have been performed and its results and metrics-even if non-significant-explicitly stated in the relevant figure legends and provided as supplementary spreadsheets.

Beyond these issues of power and reporting, the statistical methods and analytical pipelines used for the complex imaging data are inadequate and lack clarity, undermining confidence in the functional claims. A primary statistical error is the treatment of multiple cells or protrusions from a single animal as independent data points in the two-photon imaging experiments (e.g., Fig 2C, 4C) - according to the details provided in the methods section. For such data, appropriate statistical tools like linear mixed-effects models are required, yet there is no indication they were used.

This statistical oversight is compounded by significant weaknesses in the image analysis pipeline itself (MATLAB-based or JavaScript). The reliance on manual segmentation is a major concern, as it introduces subjectivity and potential observer bias, falling short of current standards that favor more reproducible, automated methods. Furthermore, the functional claims rely on a bespoke "surveillance index"-a metric based on pixel changes that has not been benchmarked against established methods or validated with ground-truth synthetic data.

5. Mechanistic Claims vs. Direct Functional Validation: The study's mechanistic claims-particularly the link between Arp2/3 loss, a reactive gene signature, and TGF- β signaling-are based on correlative data from RNA-seq (mostly) and IHC/fluorescence. While this analysis provides a valuable and intriguing hypothesis, such data alone is insufficient to prove a causal mechanism. Observing a change in a gene expression signatures via RNAseq following a genetic knockout demonstrates a correlation, but it does not establish that the genetic deletion directly causes the downstream transcriptional changes.

To elevate these findings from a correlation to a robust mechanistic causation, direct functional proof is required. This would involve experiments designed to test causality, such as a rescue experiment (e.g., re-expressing wild-type Arpc4 in the knockout background to determine if the phenotype is reversed - likely beyond the scope of the current study) or an inducible system (e.g., using a tamoxifen-inducible Cre-ERT2 to delete the gene in adult animals and demonstrate that the phenotype can be recapitulated). In the absence of this fundamental level of mechanistic validation, the proposed links, while interesting, remain speculative and are not yet sufficiently substantiated to support a causal claim.

6. Assessing Relevant Actin Structures: The manuscript uses phalloidin staining followed by FACS analysis to conclude that Arpc4-deficient microglia have reduced F-actin content (Fig. EV1B). This method is fundamentally unsuited to the biological question at hand. The central narrative of the paper revolves around the loss of dynamic, ramified protrusions. These delicate structures, which are the known primary sites of Arp2/3-dependent actin assembly in motile cells, are for sure lost during the enzymatic and mechanical dissociation required for preparing single-cell suspensions for FACS. Therefore, the resulting measurement reflects, at best, the F-actin content of the cell soma and cortical actin, not the functionally relevant actin networks within the protrusions implicated in the morphological phenotype. A far more direct and informative experiment would have been to perform F-actin staining on microglia plated in culture, allowing for direct visualization and quantification of F-actin within the lamellipodia and protrusions that the authors claim are defective.

In summary, in its present form the core claims of this manuscript are not substantiated and I don't recommend it for publication.

Response Letter to Reviewers' Comments (Point-by-point)

We are grateful to the reviewers for their thorough analysis of our submission, their comments and criticism. The resultant new data sets, data analyses, and text changes further improved our manuscript on the functional consequences of Arp2/3 depletion in microglia in vivo. We addressed each reviewer comment, including the misunderstandings of Reviewer 3 regarding the mouse model used, in the following, and have highlighted all relevant changes in yellow in the text, figure legends, and methods of the revised manuscript.

We thank all reviewers for their time in evaluating our study.

Tim Lämmermann

Director, Center for Molecular Biology of Inflammation, University of Münster, Germany

Referee #1:

The present study uncovers an unknown function of the Arp2/3 complex: preserve the homeostasis of tissue-resident microglia within the central nervous system (CNS). By genetically deleting the essential subunit Arpc4, the authors investigated the role of the Arp2/3 complex in microglia—an immune cell type that is challenging to study outside its native environment. Their findings add to the growing body of evidence suggesting that depletion of Arp subunits and disruption of Arp2/3-mediated actin branching lead to distinct cellular phenotypes, which vary depending on the myeloid cell type examined.

In this study, the authors report that microglia protrusion branching is altered, as well as motility and chemotaxis towards a wound. They also observed an increase in uptake of extracellular material, which confirms the previously described antagonism between cell migration and macropinocytosis/phagocytosis. The authors may want to discuss the potential mechanisms underlying this increase (Formin upregulation or others?).

We thank the reviewer for the very positive evaluation of our work.

We agree with the reviewer that our data confirm previous concepts on an antagonism between cell migration and macropinocytosis/phagocytosis and have included this aspect now in our discussion. Re-analysis of our bulk RNAseq data revealed indeed upregulated expression of some members of the formin/diaphanous family of proteins, in particular *Diaph1*, *Diaph3*, *Fmn1*, and *Fmnl2*, in *Arpc4*-deficient microglia (**Reviewer Figure R1**). These changes might contribute to the observed phenotypes. We have now also included these expression profiles in **Figure EV5**.

A
Reviewer Figure R1: Violin plots showing gene expression levels of formin/diaphanous family of proteins in microglia isolated from *Arpc4*^{ΔCx3cr1} and control mice. Data were retrieved from bulk RNA seq data sets from CNS-isolated microglia, analyzed in Figures 3C–F.

Referee #2:

In the manuscript 'Arp2/3 complex controls microglia dynamics and Homeostasis' Safaiyan et al. describe the effects of a microglia/myeloid cell-specific knockout of *Arpc4* on microglia morphology, motility and transcriptome. Overall, the manuscript is clearly written and contains well-designed and executed experiments and the conclusions are supported by multiple lines of evidence. I have a few recommendations for the authors to further improve and clarify their manuscript.

We thank the reviewer for the very positive evaluation of our work.

1. The role of Arp2/3 in TGF beta signaling as depicted in figure 8 could benefit from some additional analysis of the collected data. In this figure the authors demonstrate an overlap in gene expression changes between the Arpc4 KO and a Tgfr2 KO, as well as changes in SMAD phosphorylation that is dependent on the deletion of Arpc4. In the discussion it is mentioned that the SMAD data provide mechanistic insight. In order to clarify this mechanistic insight, I think it is important to address whether the relevant receptors and downstream molecules of TGFb signalling are expressed to the same degree in the KO microglia? Is the regulation at the level of the activation/downstream signaling, or is it at the level of the presence of the receptors? (similar to decreased expression of P2RY12 as shown in the manuscript).

We thank the reviewer for this constructive comment. When re-analyzing our bulk RNA sequencing data from CNS-isolated microglia, we detect that transcript levels of TGFβ receptors remain unchanged, whereas the expression of several downstream signaling molecules is reduced in microglia of *Arpc4*^{ΔCx3cr1} mice (**Reviewer Figure R2**). To our knowledge, direct detection of TGFβ receptor protein expression in microglia using flow cytometry or immunohistology has not yet been reported in the literature, and our own attempts were also unsuccessful. If the reviewer feels that we should include these data in the manuscript, we can do so.

Reviewer Figure R2: Violin plots showing the expression level of *Tgfr1* and *Tgfr2* genes (A) and genes related to SMAD molecules (B) in microglia in *Arpc4*^{ΔCx3cr1} and control mice.

2. There is an almost complete lack of overlapping clusters of microglia from wt and KO animals. This can be biological, as suggested by the authors, however it could theoretically also be the result of technical variation induced by the sampling and pooling. The methods section on this part is not very extensive making it not possible to evaluate the contribution of each. For example, to me it is not clear how many animals were included to collect the cells, if they were collected in batches or in 1 experiment and how or if they were pooled. I recommend to expand on this in the methods section, but maybe also include a small graphical overview of the setup in the figure as panel a.

We thank the reviewer for this comment and agree that providing more methodological detail will clarify the origin of the observed separation. For this experiment, we used 4 male controls, 4 male KO littermates, all age-matched and housed under identical conditions. Animal dissection, FACS-based microglia isolation, GEM generation, and cDNA amplification were performed on the same day using identical reagents, instrument settings, and a single operator. Cells from each mouse were individually barcoded using 10X Feature Barcoding, counted, and pooled in equal numbers immediately after sorting, resulting in a single pooled suspension used to generate one 10X library, thereby minimizing technical or batch effects. Cell barcodes allowed retention of sample identity throughout analysis; the female sample was excluded from downstream analysis to avoid sex-related bias. We did not perform data integration or batch correction to prevent masking of biological differences; quality control thresholds were applied equally to all samples. This single-batch, barcoded, and pooled design makes it unlikely that the absence of overlapping clusters is due to technical variation. **In the revised manuscript, we have expanded the Methods section with these details.**

3. Throughout the paper IBA1 is used as a marker for microglia. As is well known, IBA1 does not only mark microglia, but myeloid cells in general. Given the shape and location of the cells I am convinced that the analysis have been done mainly on microglia, however I feel it would be appropriate to state in the manuscript that this assumption was made, and that the data could also reflect some other myeloid populations.

We agree with the reviewer that IBA1 is not only expressed in microglia, but can also be found expressed in other myeloid cell types, such as macrophages or subsets of dendritic cells. However, there is a general consensus in the field that IBA1 serves as a reliable microglia marker, when analyzing the **parenchyme of homeostatic CNS tissue** – which is exactly the tissue location and condition that we studied in this manuscript. Microglia show a strong IBA1 signal in homeostatic CNS parenchyma, whereas macrophages outside the parenchyma display weaker IBA1 signals. Infiltration of monocyte/macrophages into the CNS parenchyma would only be expected under conditions of severe tissue damage or inflammation, which we did not induce in our study. **As requested by the reviewer, we refined our statement on the use of IBA1 as the microglia marker in the text.**

4. Overall the methodology should be expanded on to enable replication of experiments.

In response to this point, we have now provided more information in the Materials and Methods section for the following methodologies: mice, immunohistochemistry, RNAscope, intracellular flow cytometry, image processing and analysis, and scRNA sequencing.

Minor points:

- In figure 2, the GFP images are mistakenly labeled as IBA1, while it is CX3CR1 that is tagged by GFP

We thank the reviewer for pointing this out and have changed it.

- In figure 2 A,B the images representing the timeline: to me it is not fully clear what the white and green parts in these images represent. I recommend including this in the figure legend

We have adapted the figure legend for this image. Cx3cr1-GFP expression of microglia is shown in white, while one of the protrusions has been surface-rendered and 3D reconstructed with Imaris software in green to display the branch complexity of the protrusion.

- Some of the details in the figures are too small to read, eg the labels in fig 3c and 3d

We have increased font sizes for labels, where possible.

- The IF images (for example in Fig3 and 6) are a bit small, if space allows, I would recommend to increase their size

We have increased the size of these figures for a better display of the imaging data.

Referee #3:

This study addresses the important and interesting question of how the actin cytoskeleton influences the homeostatic identity of microglia. The authors have generated a large dataset, including two-photon live imaging and single-cell sequencing. However, despite the technical quality of some data, the manuscript in its current form cannot be supported for publication. The study suffers from several fundamental flaws in experimental design and interpretation that undermine the validity of its major conclusions.

We thank the reviewer for the assessment of our manuscript. We were surprised by some of the unexpected comments, which we feel are founded on some misunderstandings regarding the exactly used mouse models in this study (relating in particular to major points 1-3). Our study was conducted in close collaboration with several highly respected research groups in the microglia field, including co-

authors Marco Prinz, Katrin Kierdorf, Josef Priller, and Christian Madry. Their long-standing experience, extensive biological and technical expertise in microglia played a key role in shaping our study design, experimental execution, and data interpretation.

Major Concerns:

1. Essential Experimental Controls: The study design lacks the necessary controls to isolate the effect of Arpc4 deletion. Firstly, a Cx3cr1-Cre mouse line without the floxed Arpc4 allele is an indispensable control to ensure that expression of the Cre recombinase itself does not contribute to the phenotype. In addition, the use of the Cx3cr1-Cre knock-in line means the conditional knockout animals are, by definition, heterozygous for Cx3cr1 (Cx3cr1-Cre/+). It is plausible that even this 50% reduction in Cx3cr1 expression (haploinsufficiency) could be sufficient to disrupt the baseline motility and surveillance capacity of microglia, independent of any Arpc4 deletion. Additional experiment data comparing wild-type mice to Cx3cr1-Cre/+ mice (without the floxed allele) is required to demonstrate that the motility parameters being measured are intact. This is required to conclude that the observed phenotype is specifically due to the loss of Arp2/3.

Based on this comment, we guess that the reviewer may have misunderstood the mouse strain used in our study. The reviewer refers to the *Cx3cr1^{tm1.1(Cre)Jung}* knock-in line (**Cx3cr1^{CRE}**) (Yona et al., Immunity 2013), in which Cre recombinase disrupts the *Cx3cr1* gene, leading to its deletion. We are well aware of the limitations associated with this particular model, which is why we **deliberately avoided using it.**

In our submitted manuscript, we had made it clear that we were using the ***Tg(Cx3cr1-Cre)^{MW126Gsat}*** transgenic line – which is **not a knock-in line** – in which the Cre transgene is **not inserted into the Cx3cr1 locus.** Hence, **we do not cause any haploinsufficiency by using this mouse line.** We had referred to this mouse strain already in our initial submission with the respective citations in the Materials and Methods section (MGI:5313006, Crotti et al., *Nat Neurosci* 2014) and directly in the manuscript (Zhao et al., *eNeuro* 2019; Mroue-Ruiz et al., *bioRxiv* 2024). Previous work had shown that this mouse strain is microglia-specific in the brain parenchyme (Zhao et al., *eNeuro* 2019) and suggested less Cre-mediated effects in this strain than in inducible *Cx3cr1^{CreER}* mice (Mroue-Ruiz et al., *bioRxiv* 2024).

Of course, we had assessed the influence of potential Cre-mediated effects in this transgenic Cre line, before we started our functional studies on Arp2/3 in microglia. In pre-experiments (which had not been included in the initial submission), we had compared adult *Tg(Cx3cr1-Cre) Arpc4^{+/+}* and *Tg(Cx3cr1-Cre)*-negative *Arpc4^{fl/fl}* mice and did not find any alterations in microglia numbers, ramification and surveillance between both genotypes (**Reviewer Figure R3A-C**), confirming earlier reports on adult mice by Mroue-Ruiz et al.. Additionally, P2RY12 expression on microglia was comparable between Cre-expressing and Cre-negative animals (**Reviewer Figure R3D**). Thus, the lack of P2RY12 expression is not a cause of Cre expression and can be directly connected to the depletion of Arp2/3 in *Tg(Cx3cr1-Cre) Arpc4^{fl/fl} (Arpc4^{ΔCx3cr1})* mice (**Reviewer Figure R3D**). We also show that *Tg(Cx3cr1-Cre)* expression does not induce microglia activation (**Reviewer Figure R4A-C**).

Based on these results, for our functional studies on Arp2/3 in microglia, we used a breeding strategy that allowed us the analysis of age-matched littermate mice, comparing *Tg(Cx3cr1-Cre)*-expressing *Arpc4^{fl/fl}* (knockout) with *Tg(Cx3cr1-Cre)*-negative *Arpc4^{fl/fl}* (control) mice. By using this breeding strategy, we minimized the number of excess surplus mice that result from mouse breeding, in accordance with the strict animal ethical guidelines in Germany and 3R principles (replace, reduce, refine) on animal research.

In response to the reviewer's comment, we now include our data in Figures EV1C-F and EV2, showing that *Tg(Cx3cr1-Cre)* expression does not influence the analyzed microglia parameters and does not cause the phenotypes observed in *Tg(Cx3cr1-Cre)*-negative *Arpc4^{fl/fl} (Arpc4^{ΔCx3cr1})* mice.

Reviewer Figure R3: Microglial cell characteristics in *Tg(Cx3cr1-Cre)*-expressing and *Tg(Cx3cr1-Cre)*-negative *Arpc4*^{+/+} mice.

A, B. Representative images of IBA1⁺ microglia morphology and quantification of microglia ramification and cell territory in 12-week-old *Tg(Cx3cr1-Cre) Arpc4*^{+/+} (*Cre*⁺ *Arpc4*^{+/+}) and *Cre*-negative *Arpc4*^{fl/fl} (*Cre*⁻ *Arpc4*^{fl/fl}) mice. Scale bar: 20 μm (left) and 5 μm (right) (*n* = 4 mice per group, data are presented as mean ± s.d., two-tailed Student's *t* test, ns: not significant).

C. Quantification of microglial cell number in the cortex of 3- and 12-week-old *Cre*⁺ *Arpc4*^{+/+} and *Cre*⁻ *Arpc4*^{fl/fl} mice (*n* = 3-4 mice per group, data are presented as mean ± s.d., two-tailed Student's *t* test).

D. Representative images showing IBA1⁺/P2RY12⁺ microglia in 12-week-old *Cre*⁺ *Arpc4*^{+/+}, *Cre*⁻ *Arpc4*^{fl/fl} and *Tg(Cx3cr1-Cre) Arpc4*^{fl/fl} (*Arpc4*^{ΔCx3cr1}) mice, scale bar: 20 μm.

Reviewer Figure R4: Microglial cell activation in Cre^+ $Arpc4^{+/+}$ compared to the control (Cre^- $Arpc4^{fl/fl}$) and $Arpc4^{\Delta Cx3cr1}$ mice. Representative images of AXL and Galectin 3 expression in IBA1⁺ microglia in the cortex and corpus callosum in 12-week-old Cre -negative $Arpc4^{fl/fl}$ (Cre^- $Arpc4^{fl/fl}$) (A), $Tg(Cx3cr1-Cre)$ $Arpc4^{+/+}$ (Cre^+ $Arpc4^{+/+}$) (B), and $Arpc4^{\Delta Cx3cr1}$ (C) mice, scale bar: 20 μ m.

2. Cx3cr1 Deficiency in Functional Assays: A critical ambiguity in the manuscript's reporting leads to a potential confound in the functional assays. Central claims about impaired microglial motility, surveillance, and chemotaxis (Figures 2 and 4) are based on experiments using $Arpc4^{\Delta Cx3cr1}$ $Cx3cr1^{GFP/+}$ mice. The methods and figure legends lack the precise details of the breeding strategy and final genotypes for all experimental cohorts, leaving room for interpretation. Given standard genetic conventions, the most direct and indeed, unavoidable, interpretation is that the combined use of these two lines would result in the disruption of both endogenous copies of the $Cx3cr1$ gene. If this interpretation, which appears to be the case based on the provided description, is accurate, then this constitutes a major methodological flaw with profound implications for the study's conclusions. It has been unequivocally established for more than two decades, through extensive research in the field, that the $Cx3cr1$ gene product is critically essential for several key microglial functions. These include, but are not limited to, microglial chemotaxis, motility, and their proper infiltration into the CNS. These functions are critically dependent on the very actin-related roles that the authors propose to investigate in this study.

We thank the reviewer for pointing out this general knowledge on $Cx3cr1$ function. We can confirm that all authors of the manuscript are well aware that biallelic deficiency of $Cx3cr1$ has consequences for microglia motility and function. This is why – as explained in our detailed response to Point 1 – we did not use the $Cx3cr1^{CRE}$ knock-in mouse line $Cx3cr1^{tm1.1(cre)Jung}$. Hence, our mouse crosses using $Cx3cr1^{GFP/+}$ knock-in mice and $Tg(Cx3cr1)-Cre$ mice did not result in $Cx3cr1$ deficiency. The mouse genotypes used for two-photon imaging in our studies were: $Tg(Cx3cr1)-Cre$ $Arpc4^{fl/fl}$ $Cx3cr1^{GFP/+}$ (knockout) and $Arpc4^{fl/fl}$ $Cx3cr1^{GFP/+}$ (control).

While this information was already included in the manuscript, we agree that the genotype descriptions could have been made clearer, and **we revised the relevant sections to avoid confusion**.

3. Biological Model: Even granting concerns #1 and #2 the benefit of the doubt, the current experimental model, utilizing the $Cx3cr1-Cre$ line, presents a fundamental limitation in its capacity to address the proposed mechanisms of microglial function and maintenance. The critical aspect of this model is that the $Cx3cr1-Cre$ line initiates recombination events during the embryonic stages of development, specifically targeting microglia derived from yolk-sac progenitors. This means that any observed cellular or functional alterations in the microglia within this model are not indicative of a disruption to an already established homeostatic state in the adult brain. Since the genetic manipulation likely impacts the very formation and maturation of microglia, any observed phenotype is, at best, a manifestation of a developmental defect—an "actinopathy of development" in this context.

We agree with the reviewer on this point. Indeed, a central aim of our study was to investigate the functional role of the Arp2/3 complex in microglial biology, including its involvement in microglial development. For this reason, we deliberately chose a mouse model that allowed us to study these developmental aspects. In our initial submission, we had already stated: *“To deplete the Arp2/3 complex in microglia developing and residing in the CNS tissue, we targeted the Arp2/3 subunit Arpc4, which is a critical component of the Arp2/3 complex. Hence, we generated a conditional knockout mouse model, Tg(Cx3cr1-Cre) Arpc4^{fl/fl} (Arpc4^{ΔCx3cr1}), in which Arpc4 is deleted in brain myeloid cell populations, including microglia, from their onset of development”*.

Based on this approach, we made the unexpected discovery that the Arp2/3 complex plays a role in preventing the full acquisition of a homeostatic gene signature in microglia. **We have reviewed our manuscript again and refined wording where our approach may not have been clear. We also changed the title of our manuscript to “Arp2/3 complex controls the development of homeostatic microglia”**.

4. Statistics and Analysis: The manuscript's statistical framework has several critical deficiencies that span from experimental design to data analysis and reporting. In the methods, the authors state, "No power analyses were used to predetermine sample sizes," but justify the choice based on prior literature. This justification is insufficient. The simple fact that an error, such as the use of underpowered cohorts, is propagated through the literature does not refute that it is an intrinsic statistical flaw. An error remains an error, regardless of how commonly it is made. The use of very small animal numbers (n=3-4 per group) is therefore not statistically justified and is a major concern.

The reviewer's comment raises a very general point regarding the use of living animals in research. The use of mice in biological experiments always requires careful consideration of statistical robustness alongside the ethical responsibility to adhere to the **3R principles** – particularly the principle of **“Reduction”**, which emphasizes using the minimum number of animals necessary to obtain meaningful results.

In our study, the morphological microglia phenotype observed in the *Tg(Cx3cr1-Cre) Arpc4^{fl/fl}* line is strikingly pronounced. Given the clarity of this phenotype, which already reflects in statistically significant changes between control and knockout animals, we are confident that the use of *n* = 3–4 biological replicates is scientifically appropriate and aligns with common practice in similar studies. In our view, **increasing the number of mice would not be ethically justifiable** and would **disregard the animal ethical guidelines at our institute, aiming at minimizing the number of mice for reaching a scientific goal**.

Furthermore, the authors state that both males and females were pooled because they "did not notice any influence of sex." Given the very well-documented sexual dimorphism of microglia, a study with this low N lacks any statistical power to detect such differences, making this claim unsubstantiated. A brief statement in the methods is not a substitute for transparently reporting the statistical analysis; to properly justify pooling the data, a two-way ANOVA (genotype x sex) should have been performed and its results and metrics-even if non-significant-explicitly stated in the relevant figure legends and provided as supplementary spreadsheets.

To our knowledge, there is no evidence of sex-specific effects of the Arp2/3 complex in mammalian cell types. Consistent with this, we observed the distinct morphological phenotype in *Tg(Cx3cr1-Cre) Arpc4^{fl/fl}* mice compared to control mice in both male and female animals. In line with the **3R principles**, and particularly the goal of reducing animal use, we pooled male and female mice in some datasets.

In response to the reviewer's request, we now re-evaluate existing datasets and collected additional tissues from appropriately aged and sex-matched animals. These new data were used to generate a

complete, sex-stratified dataset, which are presenting as **Reviewer Figure R5** and included into the manuscript in **Figure EV3B**. Our data could not reveal any influence of sex on the observed Arp2/3-mediated phenotype of impaired microglia ramification and surveillance.

Reviewer Figure R5. Quantification of microglia ramification and cell territory in 12-week-old female and male *Arpc4*^{fl/fl} (control) and *Tg(Cx3cr1-Cre) Arpc4*^{fl/fl} (*Arpc4*^{ΔCx3cr1}) mice ($n = 3$ mice per group, data are presented as mean \pm s.d., two-tailed Student's *t* test, ns: not significant).

Beyond these issues of power and reporting, the statistical methods and analytical pipelines used for the complex imaging data are inadequate and lack clarity, undermining confidence in the functional claims. A primary statistical error is the treatment of multiple cells or protrusions from a single animal as independent data points in the two-photon imaging experiments (e.g., Fig 2C, 4C) - according to the details provided in the methods section. For such data, appropriate statistical tools like linear mixed-effects models are required, yet there is no indication they were used. This statistical oversight is compounded by significant weaknesses in the image analysis pipeline itself (MATLAB-based or JavaScript). The reliance on manual segmentation is a major concern, as it introduces subjectivity and potential observer bias, falling short of current standards that favor more reproducible, automated methods. Furthermore, the functional claims rely on a bespoke "surveillance index"-a metric based on pixel changes that has not been benchmarked against established methods or validated with ground-truth synthetic data.

We thank the referee for their insightful comments on our imaging experiments, including imaging and statistical analysis. We understand that aspects of our statistical reporting may have been perceived as inadequate and we are happy to provide more context to explain our experimental design and rectify these concerns. We apologize if our description of the two image analysis pipelines used in this study were confusing. We would like to clarify that two image analysis pipelines were used for high resolution confocal images (**Figures 1B, EV1E and EV3B**) and for live imaging videos (**Figures 2C and 4C**).

Confocal fluorescence imaging:

Indeed, we have indeed utilized the 3DMorph pipeline (York et al., 2018, PMID: 30627639), a MATLAB-based script relying on automatic, signal-threshold-based reconstruction, which is publically available on Github (<https://github.com/ElisaYork/3DMorph>). Since its publication, it has been cited in 116 different publications (https://scholar.google.com/scholar?cites=8811923418170118838&as_sdt=2005&scioldt=0,5&hl=en), being referenced as an example of an automated reconstruction method or being used in an original biological investigation. The only manual step in the pipeline, is a validation step, we added to test that the cells segmented in one stack are cells which are not fused or cut off in the confocal stack (quality control step after reconstruction). The pipeline was already convincingly employed by us in previous manuscripts (Gres et al., biorxiv, 2024, <https://doi.org/10.1101/2024.02.27.582183>). **In the revised version of the manuscript, we now added a more detailed part in the method section to clearly demonstrate that the segmentation is not manual with this pipeline.** Furthermore, we wanted to highlight that we reconstructed up to N=30-40 cells per animal in cortex and corpus callosum, offering us a better evaluation of cell morphometry across the microglia network. The higher throughput and

unbiased reconstruction of the 3D Morph pipeline gives a much more unbiased morphometry analysis compared to the formerly used manual reconstruction of a few microglia with IMARIS.

Live imaging videos:

We appreciate the referee's feedback that, in the initial submission, the statistical tests and p-values of the live imaging data were not explicitly stated and apologize for this lack of clarity. **This information has now been added to the relevant results section and figure legends, and further details have been provided in the Methods section** to ensure full transparency.

First, the referee also expressed concerns about potential confounders in our results due to pseudoreplication, i.e. the treatment of individual cells from a single animal as independent data. While we acknowledge this reasoning from a technical point of view, it is important to note that, as evidenced by numerous single-cell studies of microglia, a significant proportion of the variance arises from differences between cells (that in fact do not behave as identical clones), in addition to differences between animals. Our statistical approach therefore relied on a balanced sampling of both levels of biological variability, considering cells and animals. In addition, large Arp2/3-mediated effect sizes on inhibition of microglial surveillance and chemotaxis by more than 80 % combined with low variance yielded sufficient power with the employed animal numbers (<https://homepage.univie.ac.at/robin.ristl/samplesize.php>). In this regard, our experiments comply with the ethical standards of the 3Rs, using the minimum number of animals necessary to produce robust and biologically meaningful results.

Considering the reviewer's point, **we have re-analysed all multi-photon imaging datasets at the level of animal in the revised version.** To this end, cell- or event-level (chemotaxis) measures were first averaged per mouse and group comparisons (WT vs. *Arpc4*-deficient) were then performed on these means per mouse. This procedure corroborated our cell-based results, documenting significantly reduced microglial surveillance and chemotaxis in *Arpc4*^{A^{Cx3cr1}} animals (**Reviewer Figure R6**). These amended graphs have now been included in the manuscript as **Figures 2C, 2D and 4C**. This strategy avoids the use of linear mixed-effects models, as suggested by the referee - a method that would certainly be appropriate in the case of subtle changes at small numbers of animals. However, we hope that the referee acknowledges that this kind of analysis is unnecessary to support our claim, given the very potent effect of *Arpc4*-deficiency leading to near-complete inhibition of surveillance and chemotaxis.

We also clarify that the 'multiple protrusions as N' referred to by the reviewer does not reflect our individual data units. Each 'N' corresponds to either individual cells or a chemotactic event; the latter reflecting the entire coordinated response of all microglia to a laser ablation. **We have now explicitly stated this in the 'Methods' section to avoid any ambiguity.**

Second, the reviewer addressed the validity of our image analysis pipeline, particularly regarding potential biases arising from the manual selection of cells and the accuracy of parameters such as the surveillance index. Although we were surprised by this rather harsh criticism, **we appreciate the opportunity to explain our reasoning behind implementing manual steps in our otherwise automated and well-established imaging pipeline** (e.g. Madry et al., 2018a (PMID: 29290552), Madry et al., 2018b (PMID: 29382767), Roufagalas et al., 2021 (PMID: 34274460), Kyrargyri et al., 2020 (PMID: 31520551), Korte et al., 2024 (PMID: 39294491), Rifat et al., 2025 (PMID: 41124273)). The imaging pipeline that we use has been initially developed and published by the Attwell Laboratory at UCL (Madry et al., 2018a, Madry et al., 2018b), which is renowned in the field of glia research and image analysis. This method, which has since been adopted and used in publications by numerous laboratories specializing in microglia, **deliberately employs a combination of manual and automated image processing of 4D imaging stacks.** Importantly, only cells that were fully contained within the z-stack throughout the entire imaging period were included in the analysis; no cells were selected arbitrarily. This manual step was explicitly employed as a 'quality assurance' to exclude artefacts or peripheral areas of cells. The second manual intervention involved the thresholding of greyscale analog images to create binary images. Due to very heterogenous signal intensities within a cell - with weaker intensities in the peripheral processes than those close to soma, or the soma itself - this critical step was also performed manually. Automation of image processing was otherwise implemented wherever deemed sensible and feasible (see Kyrargyri et al., 2019 (PMID: 31392686) for methodological details). **Furthermore, all imaging analysis was performed with the experimenter blinded to the condition, as specified in the Methods section, in order to avoid bias.**

The use of the surveillance index reflects a quantitative correlate to capture changes in microglial morphodynamics. Analysis of surveillance or "motility" is routinely performed in many studies - also by other labs (e.g. Sipe et al., 2016 (PMID: 26948129), Stowell et al., 2019 (PMID: 31636451), Liu et al.,

2019 (PMID: 31636449), Bernier et al., 2019 (PMID: 31167136), Stoessel et al., 2025 (PMID: 40174868)) - investigating microglial behavior in slices or in vivo, with the quantification essentially corresponding to the calculations underlying our index. The surveillance index is defined as the sum of non-zero pixels in both process extensions and retractions and is calculated by determining the number of added or removed pixels frame by frame. This index is a measure of the brain volume surveilled by microglia over a given time, considering the rate of process movement as well as the total number and length of processes. Hence, changes in the surveillance index result either from a loss of process complexity or decreased motility, or from both, and the index alone does not distinguish between these possibilities. We therefore additionally analyzed the ramification of microglia, expressed as a ramification index, which is defined as the ratio of the cell's perimeter to its area (normalized to that of a circle of the same area) and depends on the cell's shape, but not on its overall size. **Together, both indices provide quantitative information about the morphodynamic changes in microglia resulting from functional impairment in *Arpc4*^{ΔCx3cr1} mice.**

Reviewer Figure R6. Microglial cell dynamics in *Arpc4*^{ΔCx3cr1} and control mice.

A. Quantification of microglial cell ramification in the two-photon live imaging of acute brain slices prepared from 12-weeks-old *Arpc4*^{ΔCx3cr1} *Cx3cr1*^{GFP/+} and *Cx3cr1*^{GFP/+} control mice ($n = 3$ mice per group, data are presented as mean \pm SEM of three mice; the ramification index of 43 single cells from each mouse was analyzed, Welch's t-test, $P=0.00319$).

B. Quantification of brain area surveyed by a microglial cell per minute (surveillance index) in the two-photon live imaging of acute brain slices prepared from 12-weeks-old *Arpc4*^{ΔCx3cr1} *Cx3cr1*^{GFP/+} and *Cx3cr1*^{GFP/+} control mice ($n = 3$ mice per group, data are presented as mean \pm SEM of three mice; the surveillance index of 51 single cells from each mouse was analyzed, Student's t-test, $P=9.69 \times 10^{-4}$).

C. Time course of directed motility quantified as the distance of microglial processes to the damaged spot ($n = 3$ mice per group, each dot represents the mean \pm SEM of three mice; in each mouse, the distance of 27-33 cells from the center of the damage was analyzed, Student's t test (unpaired), $P=0.025$).

5. Mechanistic Claims vs. Direct Functional Validation: The study's mechanistic claims-particularly the link between Arp2/3 loss, a reactive gene signature, and TGF- β signaling-are based on correlative data from RNA-seq (mostly) and IHC/fluorescence. While this analysis provides a valuable and intriguing hypothesis, such data alone is insufficient to prove a causal mechanism. Observing a change in a gene expression signatures via RNAseq following a genetic knockout demonstrates a correlation, but it does not establish that the genetic deletion directly causes the downstream transcriptional changes. To elevate these findings from a correlation to a robust mechanistic causation, direct functional proof is required. This would involve experiments designed to test causality, such as a rescue experiment (e.g., re-expressing wild-type *Arpc4* in the knockout background to determine if the phenotype is reversed - likely beyond the scope of the current study) or an inducible system (e.g., using a tamoxifen-inducible Cre-ERT2 to delete the gene in adult animals and demonstrate that the phenotype can be recapitulated). In the absence of this fundamental level of mechanistic validation, the proposed links, while interesting, remain speculative and are not yet sufficiently substantiated to support a causal claim.

We thank the reviewer for this general advice and suggestions. We would like to clarify that **our manuscript does not claim to establish a direct mechanistic link** between Arp2/3 function and the observed transcriptional changes, including the reactive gene signature or impaired TGF- β signaling. As we have stated at the end of our Discussion, we “could not yet determine the detailed molecular mechanism underlying the observed tissue phenotype, which is difficult to dissect in endogenous microglia,” given the multifaceted roles of the Arp2/3 complex in cytosolic and nuclear processes. Nonetheless, we appreciate the need to ensure that no overinterpretation arises. **We have revised**

the manuscript to more explicitly state the correlative nature of these findings and toned down any language that may have mistakenly implied a direct causal relationship.

Importantly, our data clearly demonstrate that **loss of Arp2/3 in developing microglia results in a complex and unexpected cellular phenotype that extends beyond disruptions in cytoskeletal organization.** These include:

- Downregulation of homeostatic gene expression,
- Induction of a reactive transcriptional profile,
- Enhanced phagocytosis of myelin,
- Impaired TGF- β signaling,
- And characteristic morphological changes such as reduced ramification and surveillance behavior.

Many of these findings were validated using immunohistochemistry and imaging to confirm that the transcriptional changes observed by RNA-seq translate into phenotypic alterations at the cellular level.

While we acknowledge that we cannot yet resolve the hierarchy or causality among these phenotypes, we are confident that our study highlights the Arp2/3 complex as an important upstream regulator of microglial homeostasis in the CNS. We feel this is a novel and valuable contribution suitable for publication in *EMBO Reports*. If we had managed to dissect all causalities, we would have likely sent this manuscript to a higher-ranked journal.

While rescue experiments or inducible models could be helpful to demonstrate causality, they are technically and temporally beyond the scope of this study. We agree that our current data only support a correlative relationship between Arp2/3 loss and impaired TGF β signaling. Nevertheless, our data, showing nuclear pSMAD2/3 loss, overlap with *Tgfb2*-deficient transcriptomes, and pathway enrichment, provide a strong foundation for future mechanistic dissection, which we now outline. Our findings also emphasize that alterations in actin cytoskeletal components can affect gene regulatory programs and cell identity, not just cellular morphology. This is particularly relevant in the context of emerging studies in other immune cell types (e.g., T cells, mast cells, and Langerhans cells), **which similarly highlight functions of the Arp2/3 complex that go beyond structural roles.**

Finally, we agree that *Cx3cr1*-CreERT2 mice could also be a genetic tool **to explore Arp2/3's role specifically in the maintenance of microglial identity** after development in future studies. **Our current model captures both developmental and maintenance aspects of Arp2/3 function.** That said, the inducible system also has its own limitations: tamoxifen-induced recombination efficiency can vary depending on the targeted floxed locus, sometimes leading to incomplete gene deletion. The extent of protein depletion, in particular for a heptameric protein complex like Arp2/3, also depends on both the residual protein levels and protein turnover rate, which could complicate phenotype interpretation.

6. Assessing Relevant Actin Structures: The manuscript uses phalloidin staining followed by FACS analysis to conclude that *Arpc4*-deficient microglia have reduced F-actin content (Fig. EV1B). This method is fundamentally unsuited to the biological question at hand. The central narrative of the paper revolves around the loss of dynamic, ramified protrusions. These delicate structures, which are the known primary sites of Arp2/3-dependent actin assembly in motile cells, are for sure lost during the enzymatic and mechanical dissociation required for preparing single-cell suspensions for FACS. Therefore, the resulting measurement reflects, at best, the F-actin content of the cell soma and cortical actin, not the functionally relevant actin networks within the protrusions implicated in the morphological phenotype. A far more direct and informative experiment would have been to perform F-actin staining on microglia plated in culture, allowing for direct visualization and quantification of F-actin within the lamellipodia and protrusions that the authors claim are defective.

We agree with the reviewer that detection of reduced phalloidin staining by flow cytometry is a relatively crude measurement, capturing primarily the F-actin content in the cell soma and cortical

actin. We used it as a supportive measure of global F-actin changes, and **we have adapted the text accordingly**. As observed in other cell types, a general reduction in F-actin content is frequently seen upon inhibition or depletion of the Arp2/3 complex.

Contrary to the reviewer's interpretation, the central narrative of this study focuses solely on the Arp2/3 complex as critical component for the developmental transition of microglia toward a homeostatic gene expression profile and tissue surveillance function, **an unexpected finding that expands the functional understanding of Arp2/3 beyond its classical roles**, e.g. lamellopodia formation. Given that the Arp2/3 complex localizes to several cellular compartments, including lamellipodial, cortical, perinuclear, pericentriolar and also nuclear actin, it can participate in a wide range of cytosolic and nuclear processes.

Therefore, we do not yet know, and also do not claim, that the observed phenotype is directly linked to changes in F-actin content specifically in protrusions or lamellipodia.

Moreover, simple F-actin stainings of microglia plated on artificial, non-physiological culture substrates offer limited insight into the organization of the actin cytoskeleton in vivo. Importantly, we already included data from 3D cultures, where *Arpc4*-deficient microglia display thin elongated protrusions as a consequence of lost branching over time. We believe this system partially compensates for the limitations of the FACS data, as it allows real-time visualization of actin-related morphology. Discussions with actin dynamics experts, including Roland Wedlich-Söldner and Michael Sixt, have reinforced the view that more informative approaches would include live-cell imaging of Lifeact-GFP expressing microglia to assess F-actin dynamics in real time (requiring the generation of *Tg(Cx3cr1-Cre) Arpc4^{fl/fl} Lifeact-GFP* mice) or Cryo-EM of statically fixed microglia. Both of these technically demanding experiments are beyond the scope of the current study.

Dear Prof. Lämmermann,

Thank you for the submission of your revised manuscript to our editorial offices. I have now received the reports from the two referees that I asked to re-evaluate the study, you will find below. As you will see, the referees now fully support publication of your study in EMBO reports. Referee #3 has a few suggestions to improve the manuscript, I ask you to address in a final revised manuscript. Please also provide a final p-b-p-response regarding the remaining referee points and the editorial requests below.

Editorial requests:

- Please provide the abstract written in present tense throughout.

- Please order the manuscript sections like this, using only these names:

Title page - Abstract - Keywords (up to five) - Introduction - Results - Discussion - Methods - Data availability section - Acknowledgements (please include here all the funding information) - Disclosure and Competing Interests Statement - References - Figure legends - Expanded View Figure legends

Thus, please reduce the keywords to five and include all the funding information in the acknowledgements. Moreover, please remove the sections 'Ethics approval and consent to participate' and 'Consent for publication'.

- We now use CRediT to specify the contributions of each author in the journal submission system. CRediT replaces the author contribution section. Please use the free text box to provide more detailed descriptions and do NOT provide your final manuscript text file with an author contributions section. See also our guide to authors (section 'Author contributions'): <https://link.springer.com/journal/44319/submission-guidelines#cms-Revised-submissions>

- Please remove now the referee token from the Data Availability section and make sure that the datasets are public latest upon online publication of the manuscript.

- Please change the callouts to the movies to 'Movie EVx'.

- Please check again that the number "n" for how many independent experiments were performed, their nature (biological versus technical replicates), the bars and error bars (e.g. SEM, SD) and the test used to calculate p-values is indicated in the respective figure legends (main, EV and Appendix figures). Please also check that all the p-values are explained in the legend, and that these fit to those shown in the figure. Please provide statistical testing where applicable. Please avoid the phrase 'independent experiment' but clearly state if these were biological or technical replicates. Please also indicate (e.g. with n.s.) if testing was performed, but the differences are not significant. In case n=2, please show the data as separate datapoints without error bars and statistics. See also:

<https://link.springer.com/journal/44319/submission-guidelines#cms-Figure-and-data-presentation>

If n<5, please show single datapoints for diagrams. Moreover:

- Please note that the exact p values are not provided in the legends of figures 1D, 3B.

- Please note that information related to n is missing in the legends of figures EV4 D, EV5.

- Please use our reference format (using et al after 10 author names):

<https://link.springer.com/journal/44319/submission-guidelines#cms-Reference-guidelines>

- Please make sure that all the funding information is also entered into the online submission system and that it is complete and similar to the one in the acknowledgement section of the manuscript text file. Presently, the grant '01EE2303B, the Elite Network of Bavaria (Elite Graduate Program Biomedical Neuroscience, S-LW-2016-351/2)' is missing in the submission system. Please check.

- Please make sure that each figure panel is called out separately and that the panels are called out sequentially. Presently, it seems there is no separate callout for panel 3I. Please check.

- All Materials and Methods need to be described in the main text using our 'Structured Methods' format, which is required for all research articles. According to this format, the Methods section should include a Reagents and Tools Table (listing key reagents, experimental models, software, and relevant equipment and including their sources and relevant identifiers), uploaded as separate file, and a Methods section in which we encourage the authors to describe their methods using a step-by-step protocol format with bullet points, to facilitate the adoption of the methodologies across labs. More information on how to adhere to this format as well as downloadable templates (.doc) for the Reagents and Tools Table can be found in our author guidelines (section 'Structured Methods'):

<https://link.springer.com/journal/44319/submission-guidelines#cms-Manuscript-organisation-and-formatting>

- Please add scale bars of similar style and thickness to all microscopic images, using clearly visible black or white bars (depending on the background). Please place these in the lower right corner of the images themselves. Please do not write on or near the bars in the image but define the size in the respective figure legend. Presently, several scale bars are really hard to see. Please provide bigger scale bars for these images.

- During our regular image integrity checks we noted partial reuse (overlap) between images in panels EV2B. Please check. If this is intentional, please clearly explain and state this in the figure legend.

Moreover, we observed that the microscopy images within the figure set appear pixelated under analysis. This is often a result of converting original 16-bit TIFF files to RGB format for publication. While this is not inherently problematic, it can give the impression of image alteration to critical readers. To address this, please also upload the source data for the Expanded View figures. Please upload this as one folder, grouping together all the files for each EV Figure in separate folders (and ZIPed together). The source data already uploaded for the main figures shows that there are no image aberrations in the main image set.

In addition, I would need from you uploaded separately:

I look forward to seeing the further revised version of your manuscript when it is ready. Please let me know if you have questions regarding the revision.

Best,

Referee #2:

I am satisfied with the explanations provided by the authors and their changes in the manuscript

Referee #3:

The authors have substantively addressed the major concerns from my initial review. The manuscript-killing confounds I identified-Cre-mediated Cx3cr1 disruption and statistical pseudoreplication-have been resolved through clarification, additional controls, and appropriate reanalysis.

Resolved Issues

1. Cx3cr1/Cre Controls: My concern regarding Cx3cr1 haploinsufficiency was based on an incorrect assumption about the Cre line, for which I apologize. The clarification that Tg(Cx3cr1-Cre)MW126Gsat is transgenic (not a knock-in) resolves this. The new Cre-only control data (EV1C-F, EV2) adequately demonstrate Cre expression alone does not cause the phenotypes.
2. Genotype Clarity: The explicit genotype statement resolves the biallelic Cx3cr1 loss concern.
3. Developmental Framing: Retitling to "development of homeostatic microglia" accurately reflects what the model demonstrates.
4. Pseudoreplication: Mouse-level reanalysis (per-animal averaging before group comparison) is correct and resolves my primary statistical concern.
5. Mechanistic Claims: Acknowledgment that Arp2/3-TGF- β links remain correlative is appropriate. Please use of "associated with" / "suggests" rather than causal language.

Remaining Issues Requiring Attention

A. Appeal to Authority: About the preamble citing "highly respected research groups" and named collaborators. While I have no

doubt these well known and recognized scientists are excellent and have made central contributions to our field and to the understanding of glial cell biology, experimental design and data quality must stand on their own merits. I raise this not to be contentious, but because such framing can inadvertently undermine rather than strengthen a rebuttal.

B. Sample Size Justification: Invocation of 3R principles to justify $n=3-4$ mice per group conflates two distinct considerations. The 3Rs mandate minimal animal use consistent with robust results-not acceptance of potentially underpowered designs as an ethical good in itself. Your post-hoc link to a power calculator does not constitute prospective power analysis. That said, given the dramatic effect sizes reported ($>80\%$ reduction in surveillance and chemotaxis, near-complete phenotypic separation), $n=3-4$ is likely to be adequate for these specific endpoints, but please:

- Report effect sizes (Cohen's d) alongside p -values for key comparisons
- Briefly acknowledge in Methods that sample sizes were based on expected large effects from pilot data

C. Preprint Citations: Mroue-Ruiz et al. (bioRxiv 2024) supports claims about the Cre line. Acknowledge preprint status where cited.

Comment on Statistical Approach

The authors argue that mixed-effects models are unnecessary "given the very potent effect." This combines two distinct issues. Mixed-effects models are the principled framework for nested data (cells within animals) because they correctly partition variance at each level and propagate uncertainty without information loss-not because they help detect subtle effects. Effect magnitude is irrelevant to whether the error structure is correctly specified.

Aggregation to animal-level means, as performed in the revision, is a valid simplification that avoids pseudoreplication, but it is not statistically equivalent: it discards within-animal variance information and assumes balanced design. Although no action is required for this manuscript, I note this for methodological clarity and I am of the opinion that one should encourage mixed-effects approaches as standard practice for nested designs.

Point-by-point response to editorial requests and referee points (EMBOR-2025-62025V2)Editorial requests:

- Please provide the abstract written in present tense throughout.

We have changed this accordingly.

- Please order the manuscript sections like this, using only these names:
Title page - Abstract - Keywords (up to five) - Introduction - Results - Discussion - Methods - Data availability section - Acknowledgements (please include here all the funding information) - Disclosure and Competing Interests Statement - References - Figure legends - Expanded View Figure legends

We have arranged this accordingly.

Thus, please reduce the keywords to five and include all the funding information in the acknowledgements. Moreover, please remove the sections 'Ethics approval and consent to participate' and 'Consent for publication'.

We provide 5 keywords and removed the relevant sections.

- We now use CRediT to specify the contributions of each author in the journal submission system. CRediT replaces the author contribution section. Please use the free text box to provide more detailed descriptions and do NOT provide your final manuscript text file with an author contributions section. See also our guide to authors (section 'Author contributions'):

<https://link.springer.com/journal/44319/submission-guidelines#cms-Revised-submissions>

We provide CRediT for author contributions in the online system.

- Please remove now the referee token from the Data Availability section and make sure that the datasets are public latest upon online publication of the manuscript.

We have removed this.

- Please change the callouts to the movies to 'Movie EVx'.

We have renamed this.

- Please check again that the number "n" for how many independent experiments were performed, their nature (biological versus technical replicates), the bars and error bars (e.g. SEM, SD) and the test used to calculate p-values is indicated in the respective figure legends (main, EV and Appendix figures). Please also check that all the p-values are explained in the legend, and that these fit to those shown in the figure. Please provide statistical testing where applicable. Please avoid the phrase 'independent experiment' but clearly state if these were biological or technical replicates. Please also indicate (e.g. with n.s.) if testing was performed, but the differences are not significant. In case n=2, please show the data as separate datapoints without error bars and statistics. See also:

<https://link.springer.com/journal/44319/submission-guidelines#cms-Figure-and-data-presentation>

If n<5, please show single datapoints for diagrams. Moreover:

- Please note that the exact p values are not provided in the legends of figures 1D, 3B.
- Please note that information related to n is missing in the legends of figures EV4 D, EV5.

We have changed this.

- Please use our reference format (using et al after 10 author names):

<https://link.springer.com/journal/44319/submission-guidelines#cms-Reference-guidelines>

We have changed this.

- Please make sure that all the funding information is also entered into the online submission system and that it is complete and similar to the one in the acknowledgement section of the manuscript text file. Presently, the grant '01EE2303B, the Elite Network of Bavaria (Elite Graduate Program Biomedical Neuroscience, S-LW-2016-351/2)' is missing in the submission system. Please check.

We will include this during the online submission.

- Please make sure that each figure panel is called out separately and that the panels are called out sequentially. Presently, it seems there is no separate callout for panel 3I. Please check.

Figure 3I is now called out.

- All Materials and Methods need to be described in the main text using our 'Structured Methods' format, which is required for all research articles. According to this format, the Methods section should include a Reagents and Tools Table (listing key reagents, experimental models, software, and relevant equipment and including their sources and relevant identifiers), uploaded as separate file, and a Methods section in which we encourage the authors to describe their methods using a step-by-step protocol format with bullet points, to facilitate the adoption of the methodologies across labs. More information on how to adhere to this format as well as downloadable templates (.doc) for the Reagents and Tools Table can be found in our author guidelines (section 'Structured Methods'):

<https://link.springer.com/journal/44319/submission-guidelines#cms-Manuscript-organisation-and-formatting>

We provide a separate file for Reagents and Tools table. We have also re-formatted some sections of the methods paper into a step-by-step protocol format.

- Please add scale bars of similar style and thickness to all microscopic images, using clearly visible black or white bars (depending on the background). Please place these in the lower right corner of the images themselves. Please do not write on or near the bars in the image but define the size in the respective figure legend. Presently, several scale bars are really hard to see. Please provide bigger scale bars for these images.

We have increased the thickness of all scale bars, which are all in the lower right corner of the images.

- During our regular image integrity checks we noted partial reuse (overlap) between images in panels EV2B. Please check. If this is intentional, please clearly explain and state this in the figure legend.

This was not intentional. We have now also noticed a duplication of fluorescent signals in at least one of the images of EV2B, which must have resulted from the acquisition of confocal images by tile scans, which are later stitched together. Such stitching is sometimes not 100% perfect. We have stated this now in the legend.

Moreover, we observed that the microscopy images within the figure set appear pixelated under analysis. This is often a result of converting original 16-bit TIFF files to RGB format for publication. While this is not inherently problematic, it can give the impression of image alteration to critical readers. To address this, please also upload the source data for the Expanded View figures. Please upload this as one folder, grouping together all the files for each EV Figure in separate folders (and ZIPped together). The source data already uploaded for the main figures shows that there are no image aberrations in the main image set.

We realized this as well, and were not happy about it ourselves. We now used a different file compression, which resulted in less pixelated images in the main and EV figures. We also provide the original images of EV figures as ZIPped file.

In addition, I would need from you uploaded separately:

- a short, two-sentence summary of the manuscript (not more than 35 words).

We have uploaded this.

- two to four short (!) bullet points highlighting the key findings of your study (two lines each).

We have uploaded this.

- a schematic summary figure as separate file that provides a sketch of the major findings (not a data image) in jpeg or tiff format (with the exact width of 550 pixels and a height of not more than 400 pixels) that can be used as a visual synopsis on our website.

We have uploaded this.

Best,

Referee #2:

I am satisfied with the explanations provided by the authors and their changes in the manuscript

We thank the reviewer for the support.

Referee #3:

The authors have substantively addressed the major concerns from my initial review. The manuscript-killing confounds I identified-Cre-mediated Cx3cr1 disruption and statistical pseudoreplication-have been resolved through clarification, additional controls, and appropriate reanalysis.

We thank the reviewer for acknowledging our efforts.

Resolved Issues

1. Cx3cr1/Cre Controls: My concern regarding Cx3cr1 haploinsufficiency was based on an incorrect assumption about the Cre line, for which I apologize. The clarification that Tg(Cx3cr1-Cre)MW126Gsat is transgenic (not a knock-in) resolves this. The new Cre-only control data (EV1C-F, EV2) adequately demonstrate Cre expression alone does not cause the phenotypes.
2. Genotype Clarity: The explicit genotype statement resolves the biallelic Cx3cr1 loss concern.
3. Developmental Framing: Retitling to "development of homeostatic microglia" accurately reflects what the model demonstrates.
4. Pseudoreplication: Mouse-level reanalysis (per-animal averaging before group comparison) is correct and resolves my primary statistical concern.
5. Mechanistic Claims: Acknowledgment that Arp2/3-TGF- β links remain correlative is appropriate. Please use of "associated with" / "suggests" rather than causal language.

Remaining Issues Requiring Attention

A. Appeal to Authority: About the preamble citing "highly respected research groups" and named collaborators. While I have no doubt these well known and recognized scientists are excellent and have made central contributions to our field and to the understanding of glial cell biology, experimental design and data quality must stand on their own merits. I raise this not to be contentious, but because such framing can inadvertently undermine rather than strengthen a rebuttal.

If the editor wishes, we can submit an edited version of the previous response letter (which we have not done yet).

B. Sample Size Justification: Invocation of 3R principles to justify n=3-4 mice per group conflates two distinct considerations. The 3Rs mandate minimal animal use consistent with robust results-not acceptance of potentially underpowered designs as an ethical good in itself. Your post-hoc link to a

power calculator does not constitute prospective power analysis.

That said, given the dramatic effect sizes reported (>80% reduction in surveillance and chemotaxis, near-complete phenotypic separation), n=3-4 is likely to be adequate for these specific endpoints, but please:

- Report effect sizes (Cohen's d) alongside p-values for key comparisons
- Briefly acknowledge in Methods that sample sizes were based on expected large effects from pilot data

We have adapted the Methods section accordingly, and provide Cohen's d values with p-values in separate uploaded file.

C. Preprint Citations: Mroue-Ruiz et al. (bioRxiv 2024) supports claims about the Cre line. Acknowledge preprint status where cited.

The paper had been already cited in the submitted revision of the manuscript.

Comment on Statistical Approach

The authors argue that mixed-effects models are unnecessary "given the very potent effect." This combines two distinct issues. Mixed-effects models are the principled framework for nested data (cells within animals) because they correctly partition variance at each level and propagate uncertainty without information loss-not because they help detect subtle effects. Effect magnitude is irrelevant to whether the error structure is correctly specified.

Aggregation to animal-level means, as performed in the revision, is a valid simplification that avoids pseudoreplication, but it is not statistically equivalent: it discards within-animal variance information and assumes balanced design. Although no action is required for this manuscript, I note this for methodological clarity and I am of the opinion that one should encourage mixed-effects approaches as standard practice for nested designs.

Prof. Tim Lämmermann
University of Muenster, Germany
Institute of Medical Biochemistry
Von Eschmarch Str. 56
Muenster 48149
Germany

Dear Prof. Lämmermann,

Thank you for the submission of your final revised manuscript to our editorial offices. I now went through it and your final p-b-p-response and consider the remaining concerns and suggestions of referee #3, and the editorial requests, as adequately addressed.

I am thus very pleased to accept your manuscript for publication in the next available issue of EMBO reports. Thank you for your contribution to our journal.

You may qualify for financial assistance for your publication charges - either via a Springer Nature fully open access agreement or an EMBO initiative. Check your eligibility: <https://link.springer.com/journal/44319/how-to-publish-with-us>

Yours sincerely,

>>> Please note that it is EMBO Reports policy for the transcript of the editorial process (containing referee reports and your response letter) to be published as an online supplement to each paper. If you do NOT want this, you will need to inform the Editorial Office via email immediately. More information is available here: <https://link.springer.com/partners/embo-press/editorial-policies#Peer%20review>